# Dynamic contrast enhancement and flexible odor codes

Srinath Nizampatnam[1,2], Debajit Saha[1], Rishabh Chandak [1] & Baranidharan Raman [1,2]

Sensory stimuli evoke spiking activities patterned across neurons and time that are hypothesized to encode information about their identity. Since the same stimulus can be encountered in a multitude of ways, how stable or flexible are these stimulus-evoked responses? Here we examine this issue in the locust olfactory system. In the antennal lobe, we find that both spatial and temporal features of odor-evoked responses vary in a stimulus-history dependent manner. The response variations are not random, but allow the antennal lobe circuit to enhance the uniqueness of the current stimulus. Nevertheless, information about the odorant identity is conf

ounded due to this contrast enhancement computation. Notably, predictions from a linear logical classifier (OR-of-ANDs) that can decode information distributed in flexible subsets of neurons match results from behavioral experiments. In sum, our results suggest that a trade-off between stability and flexibility in sensory coding can be achieved using a simple computational logic.

[1] Department of Biomedical Engineering, Washington University in St. Louis, St. Louis 63130, USA. [2] Department of Electrical and Systems Engineering, Washington University in St. Louis, St. Louis 63130, USA. These authors contributed equally: Srinath Nizampatnam, Debajit Saha, Rishabh Chandak. Correspondence and requests for materials should be addressed to (email: B.R. barani@wustl.edu)

The key task of a sensory system is to transduce and represent information about environmental cues as electrical neural activities so that the organism may generate an appropriate behavioral response. The precise format in which neural activities represent stimulus-specific information i.e., 'the neural code' has been a topic of great debate in neuroscience[1-6]. Both patterns of spiking activities distributed across an ensemble of neurons[7-14] (i.e., 'the spatial code'), and the temporal features of these neural spike trains, such as their synchronicity[15,16], relative response latencies[17-19], dynamics[4,20-26] (i.e. 'the temporal code') have been shown to be important for sensory coding. While stimulus-specific information exists in both spatial and temporal dimensions, what is not understood is the stability or constancy with which these neural coding schemes allow recognition of the same stimulus encountered in a variety of different ways. We examined this issue in this study.

Alternately, flexibility in sensory representation will be necessary for carrying out certain computations. Often, sensory cues encountered by an organism change dynamically, and the same sensory cue can be encountered in a variety of different contexts (for example, coffee beans in a coffee shop or in a perfume shop). Therefore, emphasizing novelty/uniqueness or suppressing/masking redundancies becomes necessary for detecting changes in the environment. Previous electrophysiological[24,27-30], imaging[31], and psychophysical[32] studies do indeed reveal such interactions exist particularly when sensory cues are dynamically encountered. Can certain aspects of the neural responses be adapted to allow the flexibility needed for adaptive computations, while at the same time maintaining stable recognition of stimulus identity?

We examined this issue in the locust olfactory system. The insect olfactory system has been widely used for studying odor coding[9,14-16,22-25,33-36]. In this system, an odor stimulus is detected by olfactory receptor neurons (ORNs) in the antenna and the transduced electrical signals get relayed downstream to the antennal lobe (analogous to the olfactory bulb in the vertebrates). In the antennal lobe, cholinergic projection neurons (PNs) and GABAergic local neurons (LNs) interact to reformat the sensory input received from the ORNs. The spatiotemporal activity patterns of the PNs are thought to represent odor identity and intensity[14,23,24,35,37,38]. In this work, we examined how the spatial and temporal aspects of neural responses are altered when a target stimulus is delivered in several non-overlapping distractor–target odor sequences. Our results suggest a surprisingly simple neural encoding solution that provides an adequate trade-off between the representational stability needed for robust recognition and flexibility needed for adaptive sensory computations.

## Results

**PN responses vary depending on stimulus history**. We began by examining how stable or variable were odor-evoked individual PN responses when the same stimulus was delivered with different stimulus histories. Variations in stimulus histories were introduced by delivering the same odorant ('target stimulus') in various distractor-target odor sequences. A 500 ms inter-stimulus interval was used to separate the two stimuli delivered. Five different odorants were used as distractor cues and two target stimuli were used in all our experiments (Fig. 1a, b). It is worth noting that the distractor odorants comprised of four odorants belonging to different functional groups (2octanol (2oct)—an alcohol, isoamyl acetate (iaa)—an ester, benzaldehyde (bzald)—an aldehyde, citral—a terpene), and a complex blend (apple). The target odorant comprised of an odorant that has been shown previously as suitable for associative conditioning (hexanol), and an innately attractive odorant (geraniol)[24].

We found that although responses of individual PNs during solitary introductions of the target stimulus were reliable across trials, they were altered when the same stimulus was delivered with different stimulus histories (Fig. 1c, d). When compared to the solitary target odor responses, the number of spikes increased for some stimulus histories (e.g. PN2 2oct-hex vs. PN2 hex) but reduced for others (PN2 bzald-hex vs PN2 hex). Further, note that increase/decrease in target stimulus response was not based on the chemical similarity between the two cues delivered in the sequence. Our results indicate a simple rule for cross-talk: if a PN responded strongly to the first cue in the sequence (i.e., the distractor) then its response to the second stimulus (i.e. the target) was likely to be reduced. Alternately, if the spiking activity reduced below baseline levels during exposure to the distractor cue, then its response to the following target increased in most cases (all comparison made with respect to the response evoked by the solitary target stimulus; Supplementary Fig. 1a, b).

Next, we examined how the variations observed at the individual PN level affected the ensemble neural representation of the target stimuli. To visualize odor-evoked population PN responses, we used a dimensionality reduction analysis. Individual PN responses were aligned with the odor onset and binned in 50 ms time windows. The spike count per time bin of each PN became a vector component, and the spike counts of all recorded PNs in that same time bin were regarded as the high-dimensional neural response. The time series of high-dimensional PN responses during the entire target stimulus presentation (4 s duration) was then projected along three principal component axes (Methods). We found that introduction of the same stimulus generated not one ensemble neural response trajectory but a family of trajectories, one for each stimulus history. Further, consistent with the earlier studies[24,34,36,38], we found that all introductions of the target stimuli generated ensemble neural trajectories that included a fast-transient response component (~1–1.5 s from odor onset) that subsequently stabilized into steady state responses. Notably, the direction of the high-dimensional vectors (indicates variation in the set of PNs activated), and the direction of their evolution (variation in spatiotemporal patterning) during the target stimulus presentation varied in a history-dependent manner. This can be clearly noted from the misalignment between the family of neural response trajectories generated during the exposure to the same target stimulus (Fig. 1e, f, Supplementary Movies 1, 2).

How distinct or separable are these responses generated by the same stimulus? To examine this, we used a supervised dimensionality reduction technique (linear discriminant analysis or LDA) that seeks to capture the differences between the ensemble neural responses evoked by the target odorant. Projecting the data along the LDA axes, we found that the target stimulus-evoked PN responses created multiple, separable clusters, one for each stimulus history with which it was presented (Fig. 1g, h, Supplementary Movies 3, 4). This qualitative observation was further quantified using a classification analysis (Supplementary Fig. 1c, d; see Methods). Therefore, we conclude that both at the individual and population levels, PN responses that are thought to mediate identification of an olfactory stimulus in the insect olfactory system can vary significantly when the same stimulus is encountered with different histories.

**Stimulus-history dependent contrast enhancement**. Are these variations in odor-evoked ensemble neural responses random? To examine this issue, we compared the neural responses evoked by solitary and sequential presentations of the target stimulus with the distractor cue (Fig. 2a, b; Supplementary Fig. 2a, b). Our

results indicate each olfactory stimulus evoked a closed-loop trajectory that evolved in a unique direction. As can be expected from our earlier analysis, the solitary and the sequential introductions of the same target stimulus generated two different neural response trajectories. As a rule, compared to the solitary target stimulus response, when presented following the distractor cue, the ensemble response trajectory shifted further away from the responses evoked by the distractor cue. This is consistent with individual and ensemble PN response analyses (Fig. 1c, d and

Supplementary Fig. 3) where we found that during sequential presentations of the target stimulus, common neurons were suppressed (i.e. responds to both target and the preceding distractor stimulus), whereas those that responded uniquely to the target odorant tended to increase.

As a direct consequence, the angular separation between neural response trajectories of the target and distractor cues increased, which can be expected to enhance discriminability between these two odorants (i.e. contrast enhancement). To quantify this, we

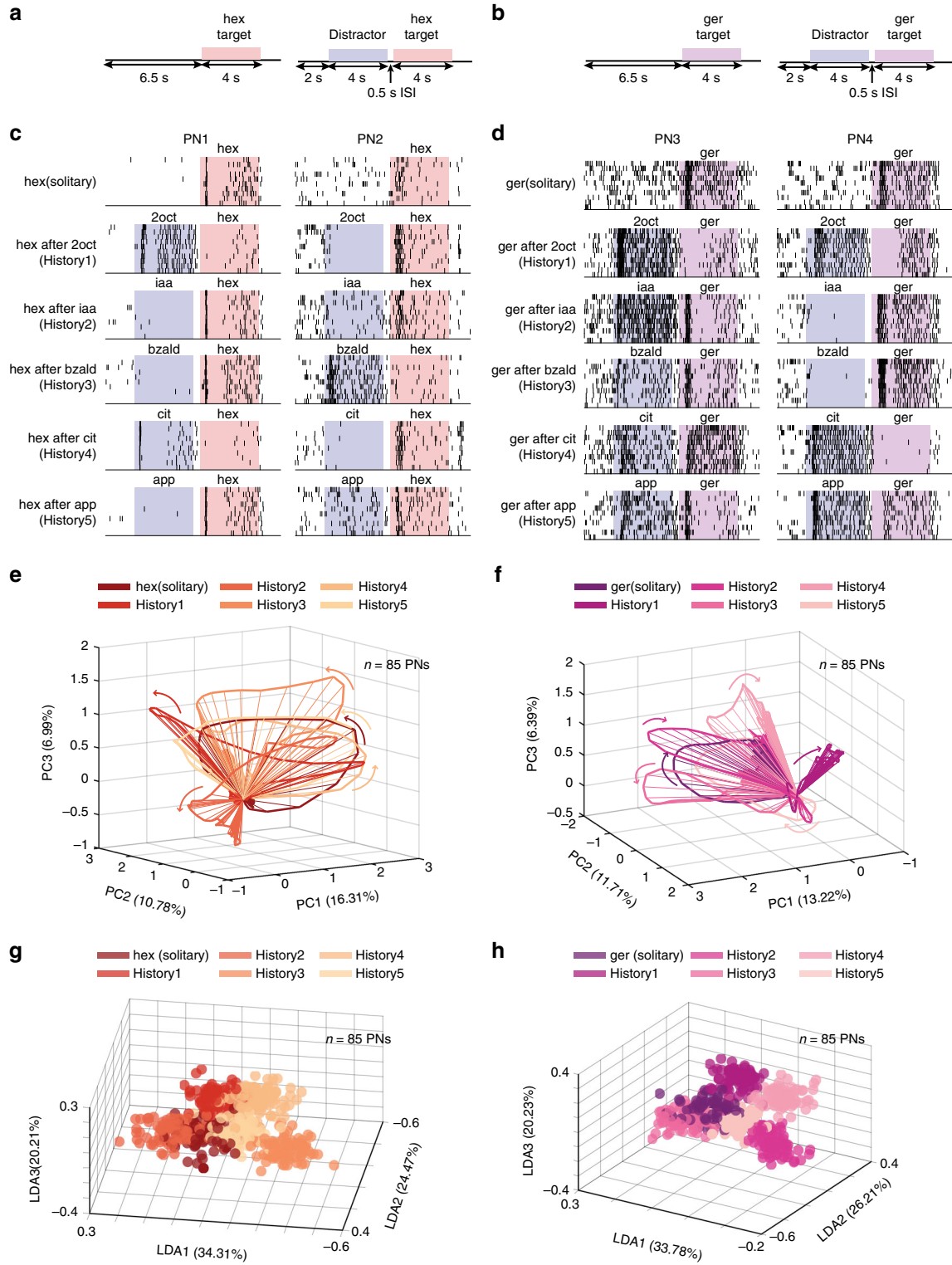

used a correlation analysis where we compared the similarity between ensemble PN activities evoked by the target stimulus and the distractor cue. We made this comparison for both the solitary exposure of the target stimulus and the sequential presentations (Fig. 2c, d; similar analysis but for later response segments shown in Supplementary Fig. 2c, d). As can be noted, the correlation between the distractor cue and the target stimulus significantly reduced during sequential presentation of the target stimulus. Further, the shift in the ensemble responses were predictable using a linear approximation where the target odor response was added with the target-minus-distractor response (i.e. uniqueness; see Supplementary Fig. 4). Hence, our results indicate that alterations in the ensemble neural responses allow the antennal lobe circuit to enhance the uniqueness of the target stimulus with respect to the preceding cue i.e., dynamic, history-dependent contrast enhancement.

**Same stimulus can activate varying combinations of neurons.** To enhance contrast with different distractor odorants, the same target stimulus must activate a different subset of the PNs. So, we wondered if there exists a unique set of neurons that is activated during all the target stimulus presentations to allow stable recognition. To understand this, we classified the PN responses into 'responsive' or 'non-responsive' categories (Methods). We found that the percentage of neurons activated and the composition of the set of responding neurons varied when the same target stimulus was presented with different histories (Fig. 3a, b). Further, the percentage of PNs that consistently responded to all introductions of hexanol and geraniol (the two target stimuli) was a smaller fraction of the set that responded to the solitary introductions of these two target stimuli.

To represent the identity of the target stimulus, the set of neurons must be consistent as well as unique. So, we examined how many of these consistent neurons were also unique responders i.e. respond to target stimuli alone (Fig. 3c, d). Surprisingly, we found that there was not a single PN that responded uniquely and consistently to hexanol in all the conditions. Only 2% neurons responded uniquely and consistently to geraniol presented with different stimulus histories. Therefore, our results indicate that combinatorial code involving a consistent and unique set of neurons may not be a suitable approach for achieving stable recognition of a sensory stimulus encountered with varying stimulus histories.

**Temporal response features also vary with stimulus history.** How robust are temporal features such as response latency and

pattern of stimulus-evoked spike trains in allowing stable odor recognition? To examine this, we defined the response latency as the first time bin when a neuron's response exceeded a certain threshold value (Methods; Supplementary Fig. 5a, b). We then classified neurons as early responders (latency <600 ms) or late/non-responders (latency >600 ms). Note that the early responders were PNs that responded before behavioral response onset (median ~600 ms[24,35]). For visualization, we represented this classification of the response latency across the ensemble of PNs as a color bar (Fig. 4a, b). The response latency vector was generated for different introductions of the target stimulus and is shown as a color bar to allow comparisons. Our results indicate that depending on the stimulus history, some of the early responding projection neurons to the solitary presentations of the target stimulus became late/non-responsive in the sequential presentations, while some of the late/non-responders became early responders. Furthermore, only a small subset of neurons consistently responded early to the target stimuli. Similar to the combinations of PNs activated (Fig. 3), the sets of consistent early responders that were also unique responders to the target stimuli were empty or near empty sets (Fig. 4c, d).

How robust are other features of PN responses such as the pattern of their stimulus-evoked spike trains? To examine this, we considered the evolution of trial-averaged PN spike counts over time (during the 4 s of target stimulus presentation) as the temporal response vector. We computed the correlation between the PN temporal response vectors generated by the same target stimulus presented with different stimulus histories (Supplementary Fig. 5c, d). The distribution of this temporal pattern similarity metric for each target-history combination revealed that more correlation values were closer to a value of 'zero' rather than 'one' (Fig. 4e, f). This indicated that the temporal patterns of PN spiking activities also change considerably with stimulus history.

Taken together, these results indicate that temporal features of PN responses are also highly variable and may not allow robust recognition of a stimulus when it is encountered with different stimulus histories.

**Robust behavioral recognition.** Our electrophysiological results indicated that combinations of neurons and their temporal response features changed when the same stimulus was presented with different histories. Therefore, we wondered whether locusts can behaviorally recognize a conditioned stimulus if it is presented in a similar fashion. To determine this, we trained locusts using an appetitive-conditioning paradigm. Briefly, during the training phase, starved locusts were presented a conditioned

**Fig. 1** Projection neuron responses vary in a stimulus-history dependent manner. **a, b** Distractor-target sequences used in this study are shown. A target stimulus (hexanol—hex or geraniol—ger) was presented solitarily or in a non-overlapping sequence with a distractor cue. All target and distractor stimuli were presented for 4 s. The time interval between the termination of the distractor stimuli and the onset of the target stimulus was 500 ms. **c** Spiking responses of two different projection neurons (PNs) in the locust antennal lobe are shown. Each tick mark represents an action potential fired by the PN. Each row corresponds to one trial and PN responses in ten consecutive trials are shown for assessing the repeatability of observed spiking response patterns. The duration when the distractor (blue) and target (red) stimuli were presented are shown using colored boxes. As can be noted, the target stimulus (hex) can be presented solitarily (top panel) or following one of the five different distractor cues: history1—2octanol (2oct), history2—isoamyl acetate (iaa), history3—benzaldehyde (bzald), history4—citral (cit), and history5—apple (app). **d** Similar plots as in **c** but showing responses of two more PNs to presentations of geraniol (ger) with different stimulus histories. **e** Responses of 85 PNs projected onto the first three principal components are shown. Each spoke represents the ensemble neural activities in a 50 ms time bin after PCA dimensionality reduction. Arrows indicate direction of evolution of the stimulus-evoked responses over time. The six trajectories shown correspond to PN responses observed following the introductions of the target stimulus either solitarily or in one of five stimulus sequences. **f** Similar trajectory plots as in **e** but visualizing ensemble PN responses evoked by introductions of geraniol (ger) with different stimulus histories. **g** Population PN responses are shown after linear discriminant analysis dimensionality reduction (n = 85 PNs). Each 3D-sphere represents an 85-dimensional PN activity vector in a 50 ms time bin. Eighty data points corresponding to ensemble neural activities evoked during 4 s of hexanol exposure with a particular stimulus history are assigned the same color. **h** Similar plot as in **g** but showing responses elicited during geraniol exposures

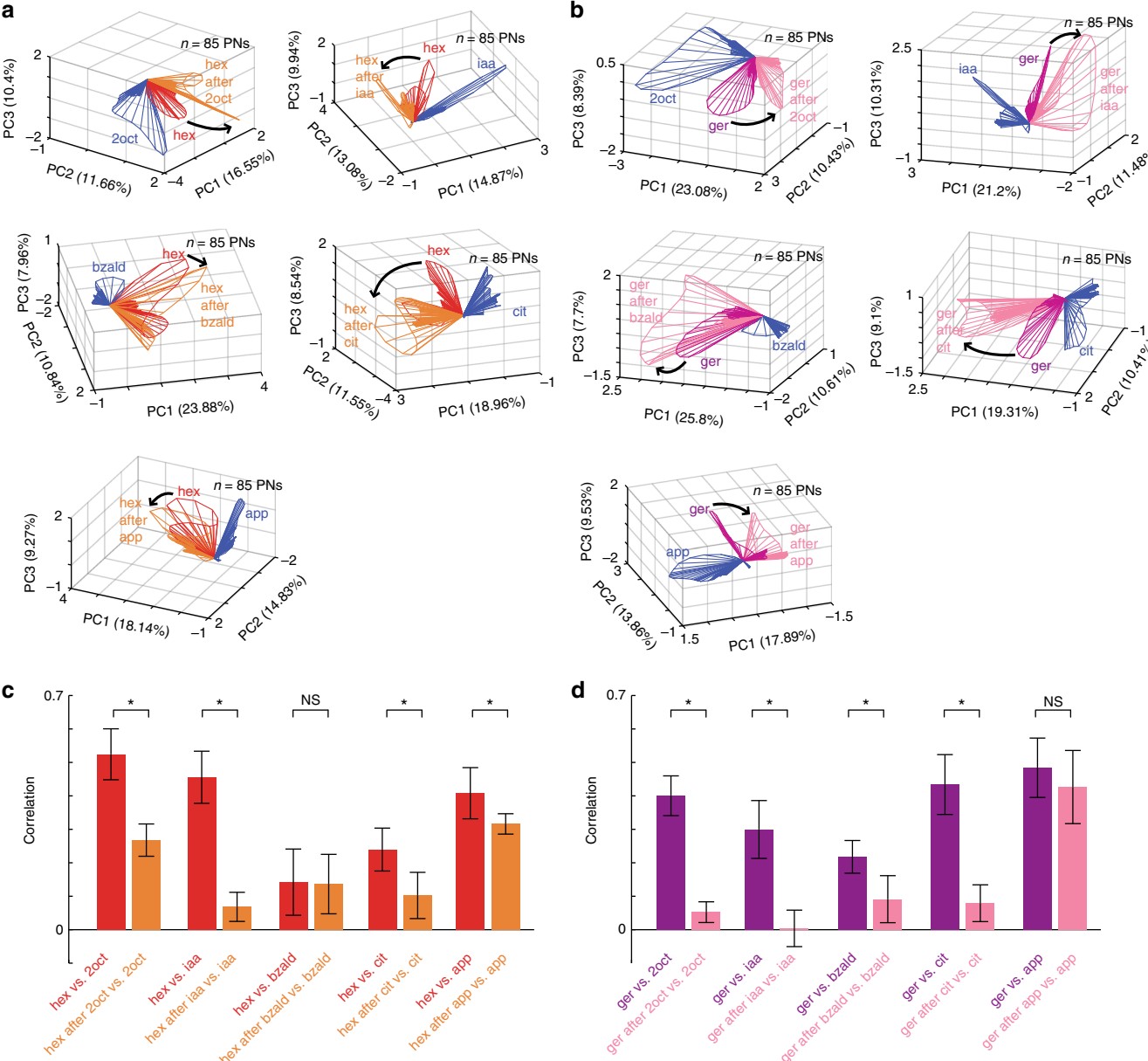

**Fig. 2** Dynamic contrast enhancement of odor-evoked ensemble responses. **a** Similar trajectory plots as shown in Fig. 1e but comparing the population PN responses generated during solitary presentations of the distractor odorant (blue trajectory) and target odorant (red trajectory) with responses elicited by the same target odorant presented after the termination of the distractor cue (orange trajectory). Five plots are shown corresponding to hexanol introductions in the same five stimulus sequences shown in Fig. 1a, c. Note that compared to the red trajectories, the orange trajectories are more distant from the blue trajectories in every panel. **b** Similar plots as in **a** are shown for five sequential presentations of geraniol. It is worth noting again that compared to the solitary geraniol responses (purple trajectories), the responses observed during sequential presentations (pink trajectories) are more distant from the distractor cue evoked responses (blue trajectories) in every panel. **c** The mean of correlation values between the ensemble PN responses ($n = 85$ PNs) evoked by hexanol and the five distractor cues are shown as bar plots. Error bars indicate ± s.d. across ten trials. The mean odor-evoked responses during the initial 1 s after stimulus introduction was used for computing these correlations. Red bars quantify similarities between solitary distractor cue and solitary hexanol presentations. Orange bars indicate similarities between the solitary distractor cue and sequential hexanol presentations. Asterisks indicate a significant decrease in the correlation (*$P < 0.05$, NS: $P > 0.05$, paired t-tests, $n = 10$ trials). **d** Similar plots as in **c** but geraniol is used as the target odorant

stimulus (hexanol; also target stimulus in electrophysiology experiments) followed by a food reward (wheat grass) with significant stimulus-reward overlap (Methods). Six such training trials were sufficient for the locusts to learn the association between the olfactory cue and the reward. In an unrewarded test phase, locusts that learned this association between the stimulus and reward opened their sensory appendages close to their mouths called maxillary palps in anticipation of the reward. The distance between the palps was tracked using a custom-written software and used as a metric for successful recognition (Methods). Notably, responses of the trained locusts to the conditioned stimulus were consistent even when the same stimulus was presented multiple times without reward during the test phase. Therefore, we assayed the palp-opening responses of the trained locusts to the conditioned stimulus presented with varying stimulus histories (Fig. 5a–f).

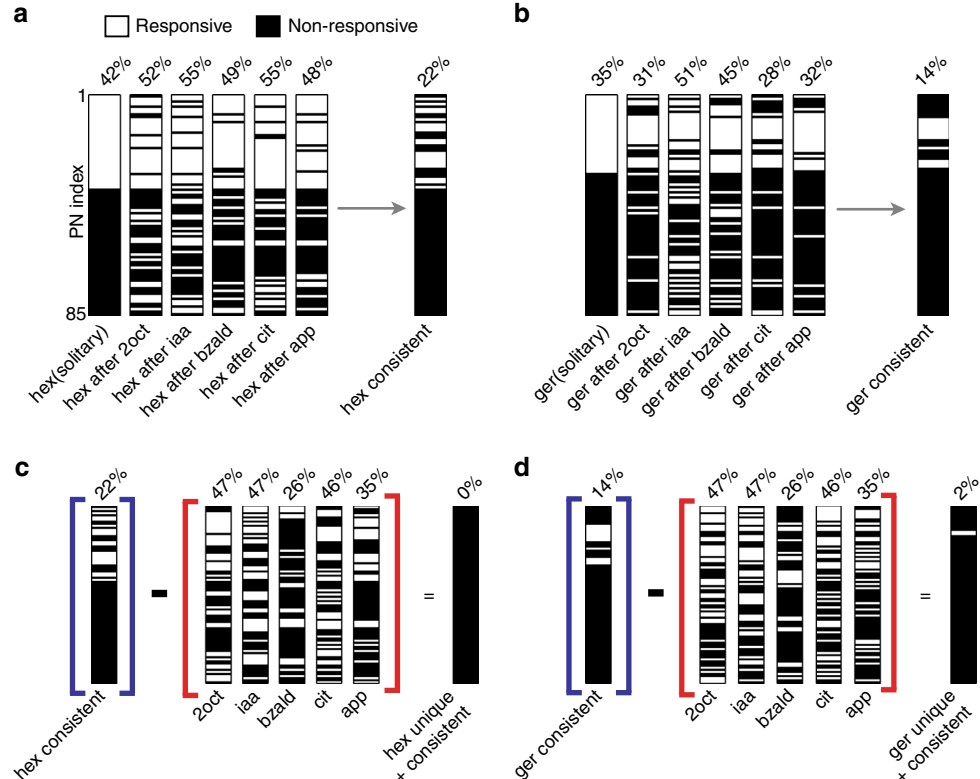

**Fig. 3** Evaluating the stability of the combinatorial code. **a** Each projection neuron was classified as being 'responsive' or 'non-responsive' to the target stimulus (Methods) and the classification of all 85 neurons are shown as a barcode. This binary classification was done for solitary and sequential introductions of hexanol. Note that all barcoded PNs are identically sorted and displayed to facilitate comparison. The percentage of PNs classified as being responsive during each hexanol presentation is shown above the barcode. The set of PNs classified as being responsive for all hexanol introductions is identified as the 'consistent set' and is also shown as rightmost barcode. **b** Similar plot as in **a** but comparing geraniol responsive and non-responsive set of PNs is shown. **c** Barcodes identifying responsive and non-responsive PNs for the five different distractor stimuli are shown. These PNs were removed from the set of neurons activated during all introductions of hexanol i.e., 'the consistent set', to identify those PNs that were both consistent and unique responders to hexanol (the rightmost barcode). As can be noted, this is an empty set for hexanol. **d** Similar plot as in **c** but identifying the unique and consistent set of PNs that responded to geraniol. For geraniol, this set consisted of only 2% of the PNs

We observed that trained locusts responded to the conditioned stimulus (hexanol) irrespective of whether it was presented solitarily or in a sequence following a distractor cue (Fig. 5g; Methods). While three of the five distractor cues did not elicit a palp-opening response (POR), two odorants (2oct and iaa) evoked POR responses that were weaker than those observed during solitary presentations of the conditioned stimulus (Fig. 5b–f, h). It is worth noting that both 2oct and iaa elicit PN responses that have considerable overlap with those evoked by hexanol[35]. So, there is some generalization of the learned POR response to these 'similar' odorants. Such olfactory generalizations have been reported in other invertebrate models[39,40] and are consistent with our prior neural/behavioral results (2oct-hex odor pair[24,35]). Evaluation of POR responses to an expanded odor panel with five additional distractor-target odor sequences indicates that the behavioral responses of locusts are indeed highly selective to the trained odorant (Supplementary Fig. 6).

In sum, our behavioral data reveal that trained locusts were able to robustly recognize the conditioned stimulus.

**Decoding with an optimal linear classifier**. Can the behavioral responses to distractor-conditioned stimulus sequences be predicted from the ensemble PN neural activities? To understand this, we first sought to determine whether the target can be robustly recognized in a distractor-target sequence (i.e. pairwise odor classification). For this approach, we used a linear optimal classifier (linear support vector machine[11,41]) to separate the target odorants from all five distractor stimuli (Fig. 6a). Surprisingly, we found that this was indeed feasible and the target stimulus can be robustly discriminated and recognized (Fig. 6b). Note that for 'n' neurons this approach would require 'n + 1' free parameters (n for the weight vector $v_{svm}$ and 1 bias term). The PN responses to target odor and all the distractor odorants must be known to determine the right set of parameters. Furthermore, even though the classification results show that robust recognition of the target stimulus is feasible from the ensemble PN activities, it still does not match well with the behavioral POR observed for these stimuli. Particularly, it is worth noting that this approach did not generate the false positives for 2oct and iaa even though these odorants generated false PORs in the behavioral assay.

**Flexible neural decoder: OR-of-ANDs**. We sought to determine the simplest approach to transform ensemble PN responses into behavioral PORs. To design this decoder, we exploited the observation that for pairs of target stimulus presentations with varying histories there was a substantial overlap in the set of neurons activated (Fig. 3a). However, when all stimulus histories were considered simultaneously, the set of consistent and unique PNs reduced significantly. Therefore, we reasoned that a decoder capable of exploiting information distributed in a flexible subset of neurons would allow robust recognition of the target stimulus.

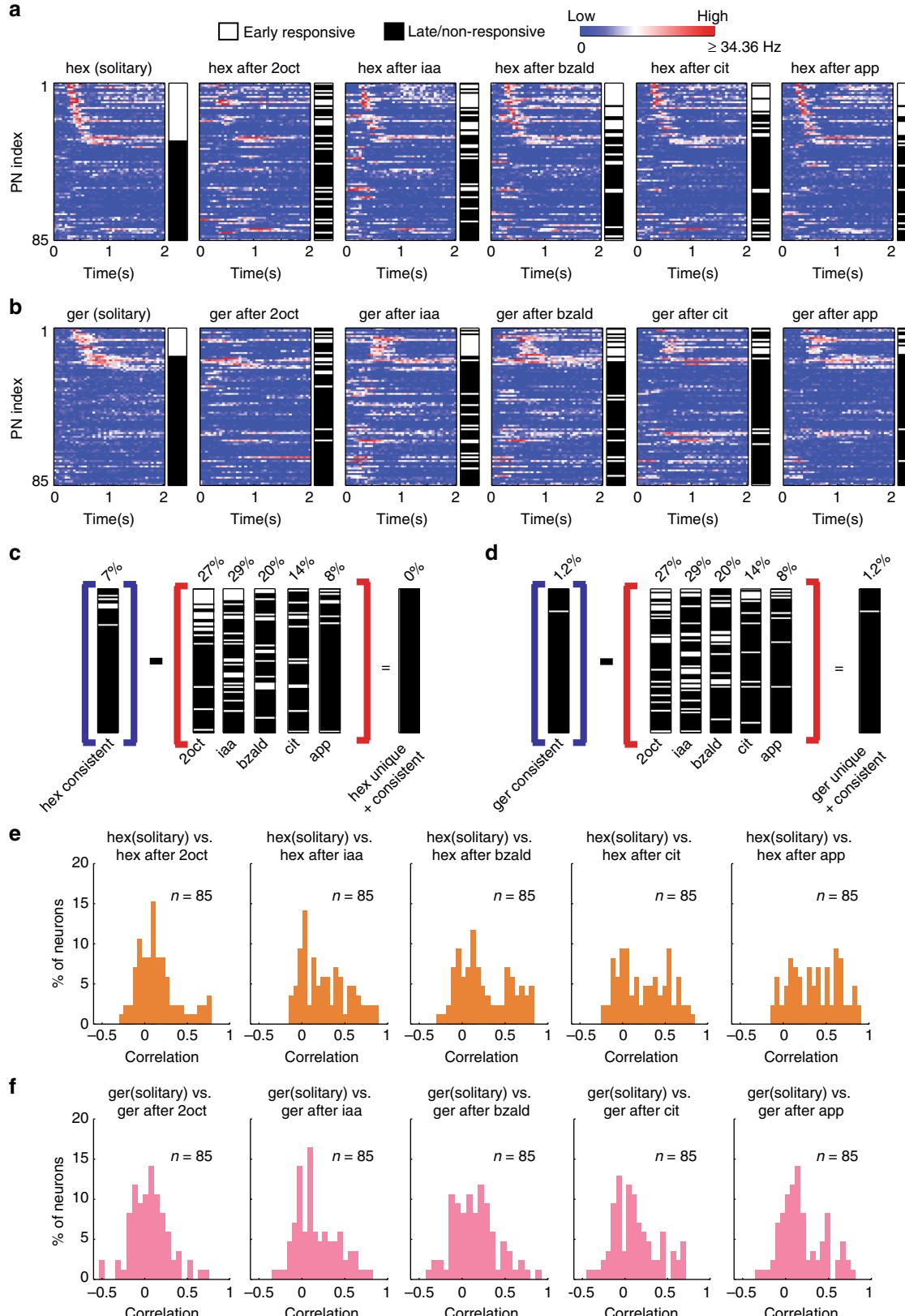

To design the flexible classifier, we summed the contributions of all neurons activated by solitary introductions of the target stimulus and disregarded the contributions of all other projection neurons (i.e. weight of '1' for each neuron activated by solitary introductions of the target stimulus, and '0' for all others). Next, the classification threshold ($m$) was set to a value less than the number of neurons ($n$) assigned a weight of '1' (i.e. $m < n$). It is worth to note that the classifier becomes analogous to an OR-of-ANDs logical operation: $m$-1 ANDs and $n$ choose $m$ ORs (or combinations) can generate an output of 'target present' from the

**Fig. 4** Temporal response features vary with stimulus histories. **a** Firing rates of 85 neurons during the 2 s following hexanol introductions are shown. A log scale was used to allow comparison between different PNs. Firing rates in 50 ms time bins were averaged across trials and are shown as a function of time. Note that the PNs were sorted based on their response latency to solitary hexanol introductions with early responders at the top and late/non-responders at the bottom. The color bars next to the firing rate plots reveal the response latency category of each PN: early responders in white and late/non-responders in black. Firing rates of the PN ensemble and the response latency vector are shown for solitary and sequential introductions of hexanol. **b** Similar plots as in **a** but now showing PN responses and their latencies to solitary and sequential introductions of geraniol. **c** Same analysis as in Fig. 3c, d. Barcodes identifying the set of PNs that responded consistently early during solitary and sequential hexanol introductions is identified. The early responsive and late/non-responsive PNs for the five different distractor stimuli are also shown. The set of PNs that were both consistent and unique early responders to hexanol is identified (the rightmost barcode). As can be noted, this is an empty set for hexanol. **d** Similar plot as in **c** but identifying the unique and consistent set of PNs that responded early to geraniol. For geraniol, this set consisted of only 1.2% of the PNs. **e** Distributions of correlation values between PN spike trains for each sequential introduction of hexanol with solitary hexanol responses are shown. **f** Similar plots as in **e** but analyzing PN responses to solitary and sequential introductions of geraniol

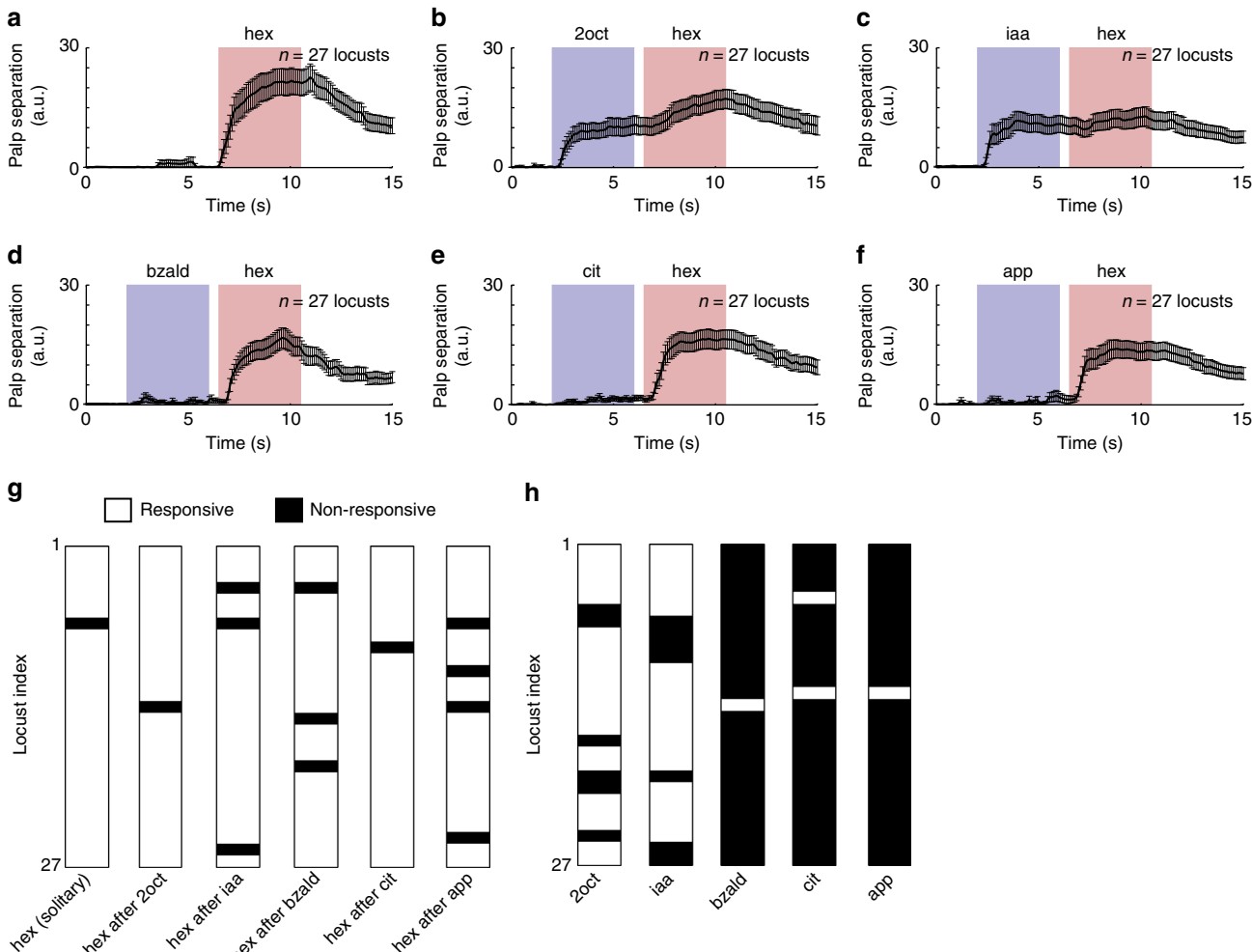

**Fig. 5** Robust behavioral response to a conditioned stimulus. **a** Locusts were trained to associate an odorant with a food reward. Trained locusts subsequently responded to the conditioned stimulus presentations by opening their sensory appendages close to their mouths called maxillary palps. The distance between the palps was tracked and plotted as a function of time. Mean palp-opening response of locusts trained to recognize hexanol (conditioned stimulus) is shown. Error bar represents standard error across locusts ($n = 27$). **b–f** Palp-opening responses to five different sequential presentations of hexanol (i.e. the conditioned stimulus) are shown. Note that these stimulus sequences are identical to the ones used in our physiology experiments. **g** PORs of each locust to hexanol were classified as responsive and non-responsive and shown for solitarily and each sequential presentation of the conditioned stimulus. **h** PORs of each locust to the five distractor odorants: 2oct, iaa, bzald, cit, and app are shown

classifier (Fig. 6c). Any $m$-out-of-$n$ neurons that respond to the solitary introductions of the target stimulus are sufficient for robust recognition of the target stimulus. Also, note that the composition of the $m$ neurons is allowed to vary across stimulus histories under this scheme.

We examined performances of both analog (PN firing rates were retained) or digital (binarized into 'responsive' and 'non-responsive') versions of this linear classifier (Fig. 6d—analog classifier, Supplementary Fig. 7—digital classifier). In both cases, the classification results from the OR-of-AND classifier revealed

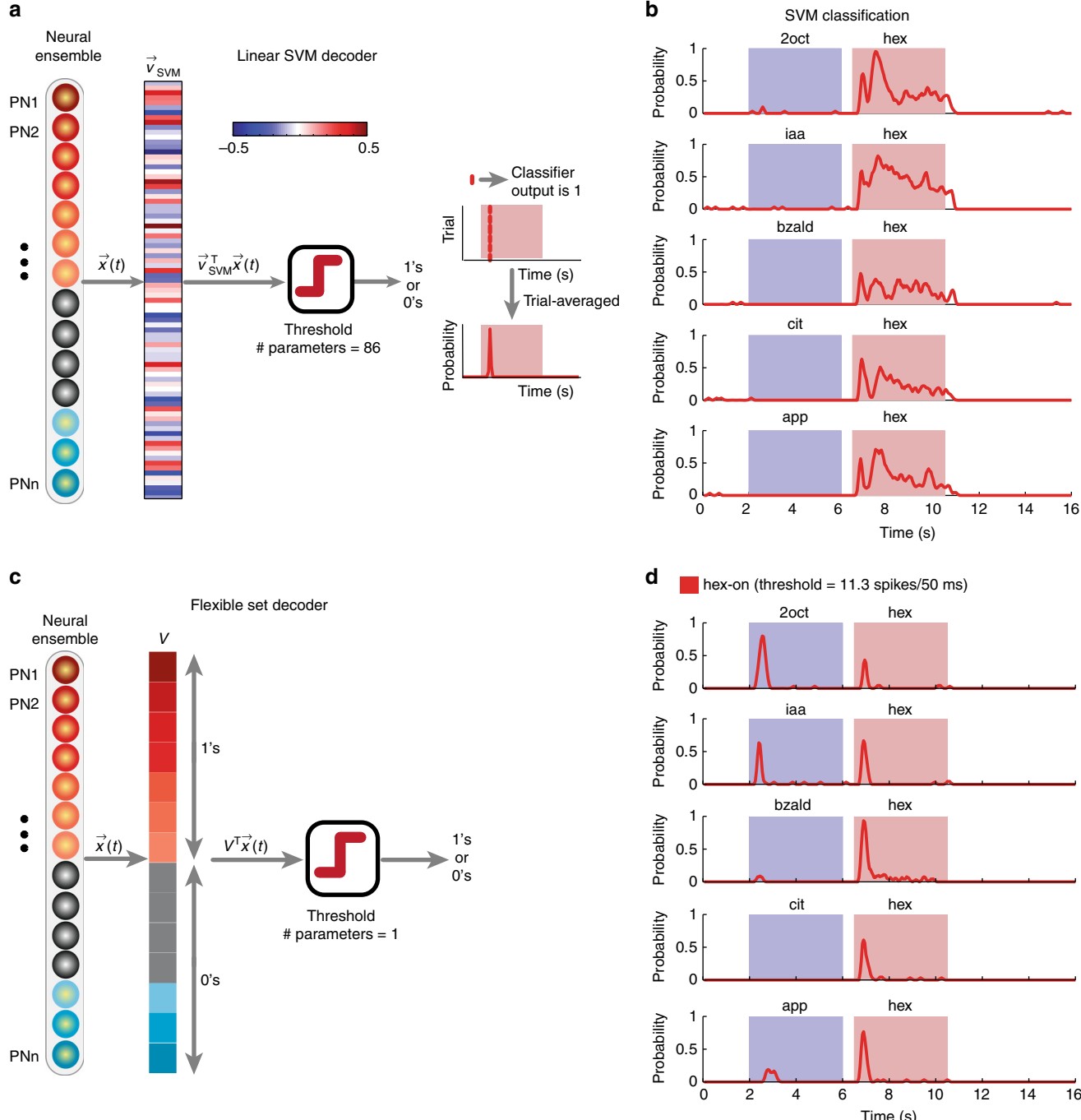

**Fig. 6** Linear neural decoding of ensemble neural activities. **a** A schematic of the multi-class linear support vector machine (SVM) classifier is shown. Here again the input to the classifier was an 85-dimensional ensemble PN activity vector in a 50 ms time bin ($x(t)$). The optimal weight vector that separates hexanol responses for all other odor-evoked and baseline activities was determined and is shown as a color bar ($v_{svm}$). Contribution of each PN was weighted based on the corresponding weight vector component ($v_{svm}^T x(t)$) and thresholded to produce a binary classification output: 'hexanol present' or 'hexanol absent'. Note that the number of parameters here are 86 (number of neurons and threshold). Probability of classification in each time bin was determined by averaging classification results across trials. **b** The probabilities of hex classification using the linear SVM classifier, across trials are plotted as a function of time for different hexanol presentations. **c** A schematic of the flexible set decoder is shown. Ensemble PN activity vector in a 50 ms time bin ($x(t)$) was the input to the classifier. The weight vector $v$ had the same number of components as $x(t)$. A vector component of '1' was assigned corresponding to PNs that were responsive to the solitary hexanol introductions, and '0' was assigned corresponding to non-responsive PNs. A dot product of $x(t)$ and $v$ resulted in summation of all hexanol responsive PNs. This scalar value was subsequently thresholded to determine the classifier output. The threshold to be exceeded was set to allow flexible subsets of neurons to contribute towards reaching the cutoff value. A digital version of the classifier when input vector $x(t)$ was also binarized and is shown in Supplementary Fig. 7. **d** Classification probabilities for the flexible set decoder are shown. Same convention as in **b**. The value of the only free model parameter, i.e., the classification threshold, is also reported

robust recognition of all presentations of the target stimulus. Interestingly, misclassifications were made for 2oct, iaa, and apple. It is worth noting that both 2oct and iaa evoked POR in locusts trained for recognizing hexanol. However, any further manipulation of classifier's free parameter (i.e. threshold or value of $m$) to reduce misclassification for apple, resulted in overall recognition performance degradation.

An earlier work showed that following termination of an odor pulse, a nearly non-overlapping set of neurons are activated(OFF ensemble neural response)[25]. These odor-evoked OFF responses are stimulus-specific. More importantly, a pattern-match with the ON neurons allowed prediction of palp-opening response and a pattern-match with the OFF neurons were better indicators of palp-closing. Therefore, we modified the decoding scheme to include two OR-of-

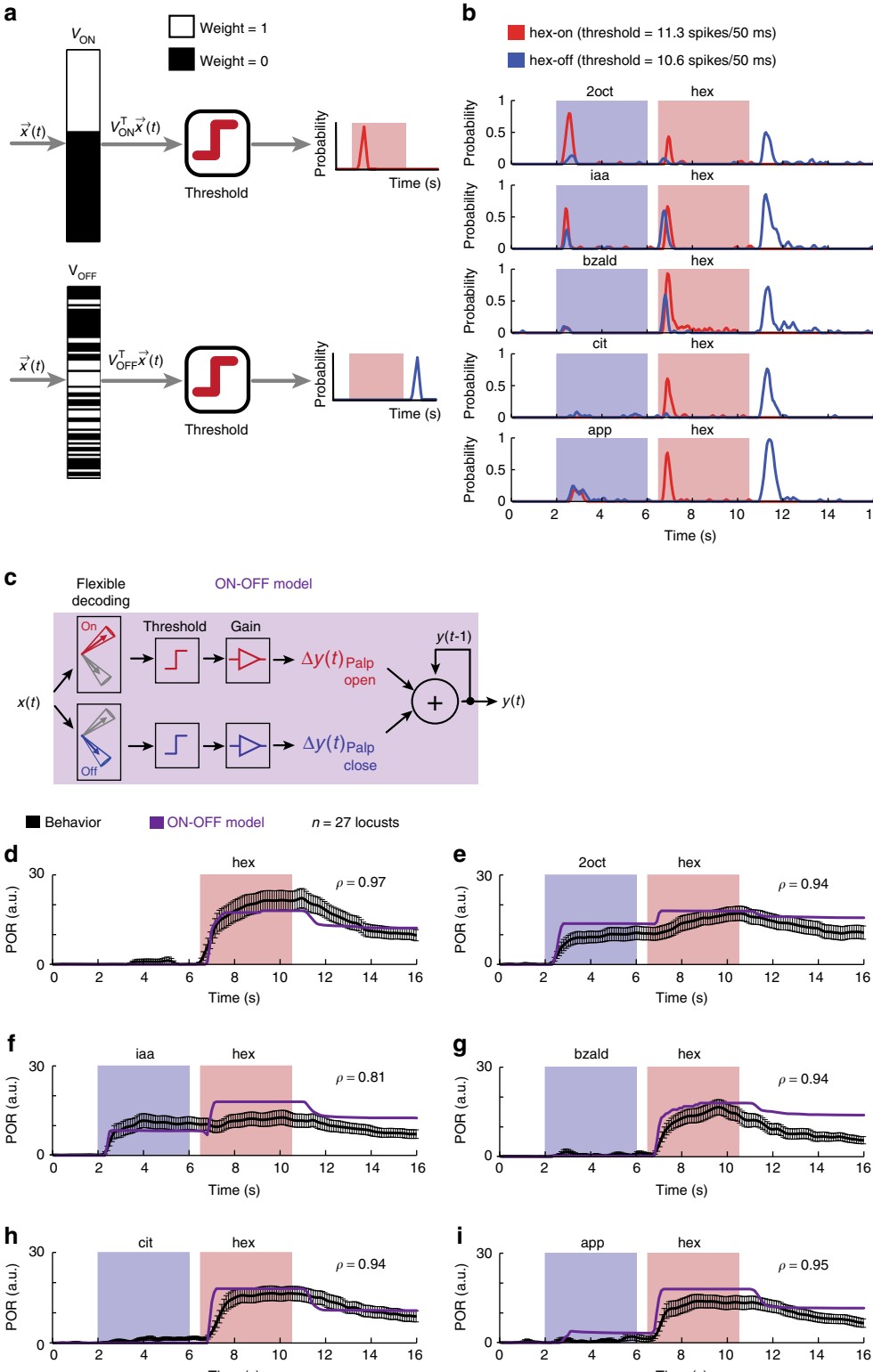

AND classifiers, one for monitoring similarity with the set of hex-ON neurons, and the other to track the similarity with the set of hex-OFF neurons. Note that the weight vectors used for the hex-ON and hex-OFF classifiers were highly non-overlapping (or orthogonal; Fig. 7a, b). Our results indicate that the performance of this ON–OFF flexible decoder (Fig. 7c–i) was comparable to the ON response model except for one important difference. We found that unlike 2oct and iaa, distractor cues that evoked POR, apple alone activated comparable pattern-matches with both hex-ON and hex-OFF neurons. When the classifier outputs were transformed into behavioral responses (Fig. 7i), these contributions canceled each other out (palp-opening vs. palp-closing). Therefore, the ON–OFF flexible decoder predicted mean POR trends that tightly matched with those observed in our behavioral experiments for all distractor-target odor sequences.

In sum, we conclude that a decoding scheme based on flexible combinations of neurons would allow stable recognition of an odorant while allowing the antennal lobe circuits to adapt their ensemble responses in a stimulus-history dependent manner.

## Discussion

What is the neural code for a sensory stimulus is a fundamental question in sensory neuroscience. Several encoding schemes that depend on a unique set of neurons activated ('the combinatorial or spatial code'), or the temporal features of neural electric discharges ('the temporal code'), or their combination ('spatio-temporal schemes') have been proposed for representing stimulus-specific information in sensory circuits. While it is well established that stimulus-specific information can indeed exist in both spatial and temporal dimensions, how robust are these schemes to extrinsic (for example, stimulus history[27,35] or ambient conditions[42]), or intrinsic (for example, plasticity[43] or internal-state, such as hungry vs. satiated[44]) perturbations? This is what we sought to determine in this study.

We examined the stability of neural representations to odorants when only the stimulus history was systematically varied. Surprisingly, this manipulation was enough to induce variations in spatial, temporal and spatiotemporal features of neural responses elicited by an odorant. Nevertheless, locusts trained to recognize a conditioned stimulus could robustly recognize and respond to the same. This mismatch between the lack of stability in the neural representation and robustness in behavior necessitated a re-examination of potential mapping schemes between neural inputs and behavioral outputs.

We used multiple decoding schemes to quantitatively examine whether robust odor recognition could be achieved when the same odorant was encountered with different stimulus histories. First, we used a linear support vector machine classifier that used $n + 1$ free parameters (where $n$ is the number of neurons) to recognize the target odorant. Results from this approach revealed that although

PN responses evoked by an odorant varied depending on the preceding stimulus, there was sufficient information to robustly recognize its identity. However, the behavioral predictions generated by this scheme mismatched with actual behavioral responses observed (Fig. 5). The false positives that we observed in the behavioral responses were not predicted using this decoding method.

Having determined that robust odor recognition was indeed possible, we next sought to determine the simplest possible decoding approach that would generate predictions consistent with the observed POR responses. For this purpose, we used a flexible decoding scheme (with ON responses only; Fig. 6d) that used only one tunable parameter. This simple scheme generated predictions that better matched with the actual behavioral responses, including the false positives. The only mismatch observed was the response to app. Although this scheme predicted a POR response to app, none were observed in the actual experimental data.

In a previous work[25], we showed that the ensemble activity during (ON response) and immediately after (OFF response) odor exposure were orthogonal and highly non-overlapping in nature. Notably, we showed that the ON responses were a better indicator of when the locusts opened their palps, and the OFF responses were a better indicator of when the locusts closed their palps. So, we combined the outputs from two flexible decoding classifiers (i.e., flexible decoding with ON and OFF responses; uses two free parameters; Fig. 7b). One used the pattern match with the ON response template of the conditioned stimulus, the other did the same with the OFF-responses elicited by the conditioned stimulus. Notably, for the app odorant, the output of the flexible ON decoder and the flexible OFF decoder canceled each other out, thereby generating more accurate predictions of PORs elicited in our behavioral assay. In sum, our results indicate that a linear scheme that decodes information from flexible subsets of neurons was sufficient to transform variable neural responses to robust behavioral outputs.

The flexible decoding approach can be easily understood if the problem is recast as one of object recognition (Supplementary Fig. 8). A simple approach to recognize an object, say a 'chair', is to first segment it into simple features (such as 4 legs, seat, back rest, and hand rest). While the presence of all relevant features (all four features in this example) allows robust recognition of the prototype chair, it may not allow generalization when other instantiations of this object are presented. Therefore, to allow generalization, it would be necessary to relax the constraint that all the relevant features need to be present. Instead, determining the presence or absence of a meaningful subset of these features, any $m$ features out of the possible $n$ features ($m < n$), can be used for achieving robustness. Higher values of $m$ allow more specificity, whereas lower values allow generalization. Our results indicate that such a decoding approach can indeed allow robust recognition of odorants presented with different stimulus histories (note that odor identity is analogous to the object to be recognized, and activation of a projection neuron is analogous to the presence of a feature). While

**Fig. 7** Predicting behavioral responses from neural activities using flexible decoding. **a** $V_{ON}$ and $V_{OFF}$ weight vectors used for generating classification results are shown. Vector component with a value '1' (i.e. responsive PN) are shown in white and the vector component corresponding to the non-responsive PNs are shown in black. Note that the set of PNs activated during and immediately after hexanol presentations are nearly non-overlapping. **b** Results from two flexible set decoders using weight vectors $V_{ON}$ and $V_{OFF}$ are shown. $V_{ON}$ was a binary weight vector with a vector component '1' corresponding to PNs responsive during hexanol presentation (i.e. stimulus ON epochs). $V_{OFF}$ was also a binary weight vector but with a vector component '1' corresponding to those PNs that were activated after the termination of the hexanol presentation (i.e. stimulus OFF period—a 4 s time window that began 500 ms after stimulus termination). Probability of classification for hex-on (shown in red) and hex-off (shown in blue) over time are shown for the odor sequences. **c** A schematic of the ON-OFF model for translating the classification results from the flexible set decoder using $V_{ON}$ and $V_{OFF}$ weight vectors into behavioral PORs. The classification probabilities (i.e. red and blue traces shown in **b**) were appropriately thresholded and scaled to generate the PORs. Note that pattern match with ON responses were used to generate palp-opening responses, whereas pattern match with OFF responses were used to generate palp-closing responses. **d–i** Observed PORs to various presentations of hex (same as in Fig. 5a–f) are shown in black. The predicted PORs generated from the ON-OFF model are shown in purple. The correlation coefficient between the actual (mean trend; i.e., the black trace) and predicted PORs are shown on each of the six panels. The model parameters were set based on PORs to solitary hexanol presentation

results presented used weight vectors with components that were either a 0 or 1, we found further improvements in classification results could be achieved by using negative weights for non-responsive neurons (not shown).

Could such a flexible decoding scheme be implemented by the insect olfactory system? To understand this issue, it would be necessary to identify the basic components of the decoder: (i) convergence of input from multiple PNs onto downstream neurons, (ii) linear combination of the inputs, and (iii) a detection threshold that does not require all input neurons to be simultaneously co-active. Existing anatomical and functional studies have shown that downstream Kenyon cells in the mushroom body linearly combine inputs from multiple projection neurons[45,46]. Further, photo-stimulation of Kenyon cell dendritic claws in fruit flies have revealed that activating more than half of these input regions is sufficient for driving these cells to spike[47] (i.e. $n/2 < m < n$ is sufficient). Therefore, anatomical and functional evidences suggest that the downstream centers in the mushroom body can indeed implement such a decoding approach.

Our results also indicate that variations in the neural responses with stimulus history were not random. Rather, the response evoked by the current stimulus was reshaped to suppress overlap with the preceding cue and increase the responses of those PNs that responded uniquely to the current cue. This simple manipulation at the level of individual PNs resulted in contrast enhancement of population neural responses. Note that contrast enhancement in olfaction happens between stimuli over time, rather than in the spatial domain as in vision.

Could some or most of the results be simply explained due to mixing of vapors from the first delivered distractor stimulus with those from the target odorant (i.e. mixture coding)? A photo-ionization detector-based characterization of the stimulus overlap indicated that at least for some odorants (for example, 2oct followed by hex case), vapors from the first odor pulse lingered longer and overlapped with the target odor presentations (Supplementary Fig. 9). As can be expected, the overlap reduced as the gap between the two pulses was increased. Notably, our results indicate that hexanol presentation with a longer delay (2 s and 10 s after termination of distractor 2oct pulses) resulted in qualitatively similar contrast enhancement as that observed for shorter delays (Supplementary Fig. 10). These results indicate that the overlap between the first and second stimulus delivered in sequence is not necessary for the contrast enhancement results reported in this study.

Nevertheless, since there is some overlap between vapors for shorter latencies, we also examined how similar or different were the processing of the binary mixtures with the odor sequences. To do this, we reanalyzed a published dataset of hex and 2oct binary mixture and 2oct-short lag-hex odor sequences (lag: 0.25 or 0.5 s). We found that a binary mixture of these two odorants evoked responses that were an additive combination of two component neural activities. Therefore, the binary mixture odor responses traced an odor trajectory in-between the two component response trajectories (i.e. green trajectory (hex-2oct mixture) was in-between the red (hex) and the blue (2oct) trajectories; Supplementary Fig. 11). However, note that for both hexanol presentations, even with 0.25 and 0.5 s lag following 2oct, the responses elicited during hex presentations were more similar to hexanol than the binary mixture or the distractor. Further, the responses shifted away from the 2oct response trajectory (i.e. magenta trajectory) indicating contrast enhancement of hex-evoked responses with respect to the preceding 2oct stimulus. Together, these results indicate that the odorants encountered in sequential fashion are processed differently than simultaneously encountered odor mixtures.

How do trial-to-trial variations compare against the changes in neural responses observed across stimulus histories? We found that individual PN responses did vary considerably across trials

(Supplementary Fig. 12a, b). However, at the ensemble level, responses were highly consistent across trials. Odor-evoked response during the early, mid and late set of trials were still aligned and traced trajectories that evolved in the same direction (Supplementary Fig. 2a, b). These results were further quantified using a correlation analysis which also confirmed that variability across trials was indeed less than those observed across stimulus histories (Supplementary Fig. 12c, d).

Can such dynamic contrast enhancement lead to other potential confounds? For example, how are PN responses altered when back-to-back pulses of the same odorant are presented? To determine this, we reanalyzed a recently published PN response dataset to two back-to-back pulses of hexanol (Supplementary Fig. 13). Our analysis revealed that in addition to a reduction in responses to the subsequent pulses[36], a few PNs that did not respond to the first pulse were activated in the second pulse, and some responding to the first pulse were inactivated. Note that the same OR-of-ANDs classifier can robustly recognize the stimulus after this subtle response perturbation.

While our results do not provide mechanistic insights on how this adaptive computing may arise, a few candidate mechanisms can easily be identified. Activity-dependent synaptic depression between the sensory neurons and projections neurons[48,49], interference due to stimulus-evoked OFF responses[25], or adaptation at the level of sensory neurons[50,51] can all contribute to the results observed in this study. Finally, while we perturbed odor-evoked neural responses using variations in stimulus histories, other factors such as ambient conditions (humidity, temperature, and air flow), presence of competing odorants, plasticity in central circuits have all been shown to introduce variations different from those that were examined here. Whether the same flexible decoding scheme proposed here can provide a generic framework for achieving trade-off between stability and flexibility remains to be determined.

## Methods

**Odor stimulation**. Odor stimuli were delivered using a standard procedure[24,25,35,36]. The following odor panel was used: hexanol, geraniol, 2octanol, isoamyl acetate, benzaldehyde, citral, apple, and cyclohexanone. All odorants were diluted to their 1% concentration by volume (v/v) in mineral oil. A carrier stream of desiccated and filtered air stream (0.75 L min$^{-1}$) was directed at the antenna to provide a constant flux. A constant volume (0.1 L min$^{-1}$) from the odor bottle headspace (60 ml sealed bottles contained 20 ml diluted odor solutions) was injected into the carrier stream using a pneumatic pico-pump (WPI Inc., PV-820) during stimulus presentations. A vacuum funnel placed right behind the locust antenna ensured removal of odor vapors. Each solitary or sequential presentation of an odorant was delivered in a pseudo-randomized manner (blocks of 10 trials) with 60 s inter-trial intervals.

**Olfactory electrophysiology**. Young locusts (*Schistocerca americana*) of either sex with fully developed wings (post fifth instar) were selected from a crowded colony. After immobilizing locusts, the brain was exposed, desheathed, and continually perfused with locust saline as reported previously[36,52]. Extracellular recordings from the antennal lobe were made using a 16-channel, 4 × 4 silicon probes (NeuroNexus). Each electrode contact pad was gold plated such that its impedance was in the 200–300 kΩ range. Raw extracellular signals were filtered between 0.3 and 6 kHz and amplified at 10 k gain using a custom-made 16-channel amplifier (Biology Electronics Shop; Caltech, Pasadena, CA). Raw data were acquired at 15 kHz sampling rate using a custom LabView data acquisition program.

**PN spike sorting**. For spike sorting, we used a conservative approach that was described in earlier works[53]. We used the following criteria for the single-unit identification: cluster separation >5 noise s.d., the number of spikes within 20 ms <6.5%, and spike waveform variance <6.5 noise s.d. Using this approach, a total of 85 PNs were identified from 9 locusts (11 antennal lobes).

Similarly, 104 PNs were identified from 12 locusts (19 antennal lobes) for the data shown in Supplementary Fig. 10.

**Dimensionality reduction analyses**. We used two-dimensionality reduction techniques for visualizing ensemble PN responses: principal component analysis (PCA) and linear discriminant analysis (LDA). For both analyses, spiking activity of each PN was binned in 50 ms non-overlapping time bins and averaged

 

across trials. Spike counts of all recorded PNs were aligned with respect to the stimulus onset and concatenated to form an 85-dimensional PN response vector per 50 ms time window. Therefore, for each odorant, we obtained a response matrix of the following dimensions: 85 neurons (rows) × 80-time bins (columns; 4 s of activity).

To perform the PCA analysis shown in Fig. 1, we first concatenated the responses of the 85 PNs to solitary and sequential presentations of hexanol or geraniol (85 PNs × 480 time bins [6 presentations × 80 time bins]). Response covariance matrices (85 × 85) were computed for these concatenated data matrices. Each 85-dimensional PN response vector was then projected onto the three principal eigenvectors (for visualization).

PCA trajectories shown in Fig. 2 were computed using a different response covariance matrix. Here, the PN ensemble responses to three stimuli were concatenated to form the data matrix: solitary target stimulus (hex or ger; 4 s), solitary distractor stimulus (4 s), target stimulus after distractor stimulus (4 s of hex or ger exposure). This resulted in 85 PNs × 240 time bins data matrices [3 stimuli × 80 time bins].

For generating figures shown in Figs. 1, 2, PN responses after PCA dimensionality reduction were linked in the temporal order of occurrence to create odor trajectories. The pre-stimulus baseline activity in the first time bin was subtracted before plotting each response trajectory. The odor trajectories shown were also smoothed using a three-point moving-average, low-pass filter.

LDA dimensionality reduction[41] analysis was also performed by first concatenating all high-dimensional PN response vectors that needed to be visualized. The projection vectors were determined such that they maximized separation between responses of different stimuli while at the same time also reducing variance within responses generated by a single stimulus.

**Confusion matrix calculations**. To quantify variability across stimulus histories, we performed a classification analysis with a leave-one-trial-out cross validation (using the 85-dimensional PN responses). The test trial to be classified was assigned the same class label as the nearest cluster centroid (calculated using training samples that included hex or ger responses for different stimulus histories). The classification results are then shown as a confusion matrix in Supplementary Fig. 1c, d.

**PN response categorization**. For analysis in this work, we classified projection neurons as 'responders' if the spike counts in any time bin during the stimulus presentation exceeded mean + 6.5 s.d. of pre-stimulus activity (2 s window just before onset of any stimulus). All PNs that did not meet this threshold were regarded as 'non-responders'.

**Response latency for PNs**. We defined PN response latency as the first 50 ms time bin when the criterion for the responsive neuron was met (i.e., spike counts exceed mean + 6.5 s.d. of pre-stimulus activity). For sequential presentations, the response latency was determined as the first time bin after the derivative of PSTH became positive (used in Fig. 4 and Supplementary Fig. 5).

**Support vector machine (SVM)**. The PN response patterns to solitary introductions of hexanol, 2oct, iaa, bzald, cit, and app were used as training samples. We regarded this as a binary classification problem, where hex responses were considered as one class, and all other responses were regarded as the second class.

The separating hyperplane was found using an SVM classifier that maximizes the functional margin between the hyperplane and the classes[41].

$$\gamma_i = y_i(\mathbf{v}_{svm}^T \mathbf{x}_i + b),$$

where $\gamma_i$ is the functional margin for the data point $\mathbf{x}_i$ [85-dimensional PN spike counts in a 50 ms time bin], and $y_i$ is the true class label for the data point [1 for hex-ON and −1 for the rest]. The weight vector $\mathbf{v}_{svm}$ (also 85 dimensional; i.e. 85 free parameters to tune the model) plus the bias term $b$ (86th model parameter), was calculated by solving the following optimization problem:

$$\min_{\gamma, \mathbf{v}_{svm}, b} \frac{1}{2}||\mathbf{v}_{svm}||^2 + C\sum_{i=1}^{m}\mu_i,$$

$$\text{s.t.} \quad y_i(\mathbf{v}_{svm}^T \mathbf{x}_i + b) \geq 1 - \mu_i \; i = 1, \ldots, m,$$

$$\mu_i \geq 0, i = 1, \ldots, m,$$

where $m$ is the total number of data points in both the classes and $\mu_i$ is the slack variable that help us avoid overfitting while solving this optimization problem. This was done using fitcsvm in MATLAB by taking the box constraint to be 0.01. After determining the optimal weight vector $\mathbf{v}_{svm}$, we classified response patterns generated during sequential stimulus presentations as follows:

$$h(\mathbf{x}(t)) = \text{sign}(\mathbf{v}_{svm}^T \mathbf{x}(t) + b),$$

where $\mathbf{x}(t)$ is the 85-dimensional PN spike counts in a 50 ms time bin, the sign function results in '1' when the input is classified as 'hexanol present' and '0' corresponded to 'hexanol absent.' (used in Fig. 6a, b)

**Behavior experiments**. Behavioral experiments were performed using locusts of either sex. All locusts used were from a crowded colony that was kept on a 12 h day–12 h night cycle (7 am–7 pm day). All behavioral experiments were performed between 10 am and 3:30 pm.

Locusts were starved for 24 h prior to their use in the appetitive-conditioning assay. The protocols that we used in this study for training locusts and tracking their palp movements were identical to an earlier study[35]. The odor delivery setup and the stimulus sequences used were identical to that described for the electrophysiology experiments.

Hexanol diluted in mineral oil (hex 1%) was used as the conditioned stimulus for all behavioral experiments in this study. Wheat grass was used as the unconditioned stimulus. During each training trial, the conditioned stimulus was presented for 6 s. Food reward was given 3 s after the onset of the conditioned stimulus. Food was given manually for the duration of ~10 s. The training phase included a total of six training trials with a 10-min interval between successive trials. Only locusts that accepted food reward in at least four out of the six training trials and had a palp-opening response (POR) in at least three training trials were regarded as 'trained locusts' and retained for the testing phase (75% of the locusts (30/40) fell into this category). The imaging software was not able to robustly track the palp movements of three of the trained locusts (3/30) and, therefore, these locusts were excluded from all analyses.

In the unrewarded testing phase, locust PORs were evaluated using six test trials. The first trial in all tests was the solitary presentation of 4 s of hexanol. This was done to ensure that trained locusts were able to recognize the solitary presentations of the trained stimulus (26/27 locusts had a strong POR to solitary hexanol). In the subsequent five trials, the conditioned stimulus was presented in non-overlapping two odor sequences used in electrophysiology experiments. The two odor stimulus sequences were presented in a pseudo-randomized manner. The inter-trial delay between test phase trials was set to at least 20 min.

Similar protocol was used for collecting the data shown in Supplementary Fig. 6. We tested for 5 additional distractor odors: neem oil, cyclohexanone (chex), geraniol (ger), L-carvone (L-carv), and methyl salicylate (m-sal). A total of 20 locusts (out of 30) met the training criteria described above.

**Locust response categorization**. For analysis in Fig. 5g, h, we classified a locust to be 'responsive' when the POR to an odor presentation was 6.5 s.d. above pre-stimulus baseline response. Further, the POR during odor presentation had to remain >20% of its peak value for at least 1 s. All the other cases were marked as 'non-responsive'.

**PID experiment**. We used a fast photo-ionization detector (miniPID, Aurora Scientific) to characterize the dynamics of stimulus delivered. Raw data were amplified (gain = 5) and acquired at 15 kHz sampling rate using a custom LabView data acquisition program (Supplementary Fig. 9).

**Data availability**. The data and code used in this study can be made available on reasonable request to the authors.

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

## Acknowledgements

We thank Dirk Albrecht (Worcester Polytechnic Institute), Shinung Ching Lab (Washington University in St. Louis), Keith Hengen Lab (Washington University in St. Louis), Nitin Gupta (Indian Institute of Technology, Kanpur), and members of the Raman Lab (Washington University in St. Louis) for feedback on the manuscript. This research was supported by an Office of Naval Research grant (N00014-16-1-2426) and an NSF CAREER grant (#1453022) to B.R.

## Author contributions

B.R. conceived the study and designed the experiments/analyses. S.N. did all the analyses, generated all the figures and wrote the first draft of the manuscript. D.S. and S.N. performed the electrophysiological recordings. R.C. did the behavioral experiments. B.R. revised the paper taking inputs from all the authors. S.N., D.S., and R.C. are equally contributing first authors.

## Additional information

**Competing interests:** The authors declare no competing interests.

