## [Peer Review File · Nature Communications]

Reviewers' comments:

Reviewer #1 (Remarks to the Author):

General Comments:

Understanding the neural coding of sensory stimuli is a key aspect in understanding how the nervous system functions. In the field of olfaction, although there have been many studies investigating the neural representation of odors across neural population and time, it is not clear how stimulus identity may be encoded across contexts, while flexibility is allowed to represent the richness of the contexts. In this manuscript, the authors attempted to address this question by investigating the population coding of target odorants among 85 locust antennal lobe olfactory projection neurons (PNs), preceded with 5 distractor odorants from different functional groups with a 500 ms interval.

They found that the PN responses to the same target odorant varied depending on the preceding distractor odorant. Moreover, the presence of the preceding distractor odorants made the PN responses to the target odorant more distinct from the distractor odorants presented alone (contrast enhancement). Intriguingly, they found only about half of the PNs responding to the solitary target odorants consistently responded across stimulus history and they did not uniquely respond to the target odorants. The temporal evolution of the PN responses to the target odorants were also highly variable across different stimulus histories.

Given this divergence of PN encoding of the target odorant depending on distractor identity, would the animal recognize it as the same odorant. The authors performed palp conditioning experiments to investigate, followed by modeling exercises to suggest that a flexible OR of ANDs decoder was able to predict the behavioral responses from the PN neural data.

The study described in this manuscript was significant, experimental design was novel, experimental data was extensive, and statistical analysis and modeling approaches were sound. It also raises interesting questions for future exploration. Please see below for major and minor issues.

Specific Issues:

Major:

1) Palp opening response is extremely reduced as a measure for odor identity. How specific is the POR to hexanol (Figure 5)? How often do the palps open for odorants other than the CS+? For example, in Figure 5 panels b and c, as mentioned in the text, the maxillary palps were already open in response to the distractor odorants. It is not convincingly shown that the locusts actually recognized the target odorant, especially in panel c. Can the authors test more odorants to demonstrate the POR is specific and reliable? This needs to be discussed further.

2) The modeling results matched the data well, but may or may not be the actual algorithm employed by the brain. Here the modeling is a proposal, not a fact. Therefore, the final sentence in the abstract should be a suggestion rather than a declaration.

Minor (presentation, typo):

1) Figure 1e, f: need to streamline the trajectories. Very difficult to distinguish them with all the spokes and 6 different shades. Supplementary videos or MATLAB figures might help.

2) Figure 1g, h: Make the spheres less opaque so the front ones do not completely block the back ones. Again, supplementary MATLAB figures would help so readers can view the plots from varied angles.

3) Supplementary Figure 1a-d: do not address the response rule as described in the second paragraph of Results on page 2.

4) Figure 5 (a) legend on page 11: second line missing "to" after "subsequently responded"

5) Supplementary Figure 4 is mistakenly titled "...5"

Reviewer #2 (Remarks to the Author):

In this manuscript, the authors describe recordings from the second layer of neurons (PNs) in the locust olfactory system, while delivering single or pairs of odor stimulus pulses. They claim that the PN representation of the second odor pulse (the target) was altered by the specific odor delivered as the first pulse (the distractor) so as to increase the contrast between target and distractor. The authors then try to explain the changed response to the target. These are the most interesting parts of the manuscript to me. The authors then use conditioned responses as a behavioral readout of odor identification and are able to predict performance from alterations in PN responses to the target odor due to the distractor. Overall, the explanation of the history-dependent changes in PN responses to the target odor represents an important addition to the field, but the manuscript requires a few important revisions.

Major comments:

1. Explaining the changed responses to the target

The most interesting part of the manuscript is the attempt to explain the changes in responses to the target odor based on two general observations.

General rule of target response being inversely proportional to distractor response

o This phenomenon should be properly quantified. Please plot new versions of Supp. fig. 1 a, b in main figure to make this point as follows: spike counts (target after distractor - target alone) v/s (distractor - baseline) to show the general rule

o The observation seems anecdotally true, but here are some examples where it appears not to be. Hence, please quantify thoroughly.

In fig. 1 c, d

§ hex after app in PN2

§ ger after cit and ger after app in PN3

§ ger after cit and ger after app in PN4

- Contrast enhancement (fig. 2, fig. 4e, f).

The authors should try to use the responses to target alone and distractor alone to predict the response to target after distractor.

For example, a reflection of the distractor representation in N-d space, could be calculated as target after distractor prediction = target vector + (target vector - distractor vector). How well does such a simple estimation match the observed responses to target after distractor?

Also, if the changes do indeed reflect contrast enhancement, the activity pattern evoked by target after distractor should be similar to target alone but more different from distractor alone. The authors should show that the correlation of target after distractor with target alone remains higher than with distractor.

If the authors are able to concretely and quantitatively show the basis of the changed responses to the target odor due to presentation of the distractor, it would really strengthen the conclusions of the paper.

2. Reduce the number of decoders used:

The manuscript uses five different decoders on the PN data collected. The authors should try to reduce the number and complexity of the decoders used, keeping in mind the useful biological property illustrated by each one. The decoders used in fig. 6 and supp. fig. 4d for example, are both trying to make the point that the PN recordings contain enough information to identify the target odor, even after the changes caused by the distractor. One should suffice.

I feel the kernel-based decoder described in supp. Fig. 4 could be dropped or shifted mostly (text) to the supplement. This would be less distracting for the reader, especially since the text refers only to supp. figures.

Also, some clarifications are needed about some of the decoders

- The thresholded OR-OF-AND classifiers described in figs. 6 and 7: How do the POR prediction traces look for the on and off decoders individually? How different are the on and off weight vectors? What fractions of the PNs have weights of 1 as opposed to 0 for each solitary odor?

- Eqn 2 in the methods section does not adequately explain how the weight vector is decided for the SVM decoder. Please explain.

- In general, the authors should be more clear about what measure of neuronal response is used as the input to each classifier. In some cases, trajectories are used, but in others, just response amplitude values (I believe - please specify more clearly). It is somewhat jarring for the reader to switch between analyses without more elaboration. For instance, how was the single response amplitude value calculated for use in the LDA plots in fig. 1? The LDA based trial classifier needs some clarification. What was the shape of the matrix on which the LDA was performed? How exactly were the 85 dimensional PN response time-series trial data binned and concatenated in order to perform the LDA? Also, how were the confusion matrices in supp. fig. 1 computed? Were distances to centroid calculated in 85 dimensional LDA component space or just the first three? They should be calculated in the full, 85-d space. I ask this in relation to the fact that all the on-diagonal classification scores are quite low and in fact, not different from background in the case of hex or ger alone or hex or ger after app. In fig. 1 gh, the LDA plots have single points as opposed to the trajectories in fig. 1 ef. How are these single points calculated? Are they trajectories or is there one point for each trial?

3. Recommend tighter focus on a central message in the manuscript

fig. 3 does not add value to the manuscript. Combinatorial coding in PNs is well established. The lack of unique, consistent responders is unsurprising. This is especially the case given that the authors tested a limited, arbitrarily chosen number of distractor odors. It can be argued that given enough distractor odors, it would be impossible to find a unique responder. I recommend moving this section to the supplement.

The section on 'Temporal response features also vary with stimulus history' is not sufficiently supported by the analysis in fig. 4 and does not really add to the central theme of the manuscript. To really make the point about changing temporal trajectories, a clearer analysis than the slightly biased correlation distributions would be needed. I suggest removing this section from the main manuscript. If the authors want to make this point, they could consider quantifying deviations of the mean target after distractor trajectory from the mean target alone trajectory and comparing those with variability in target alone trajectories. Also consider overlaying a trace for the distribution of correlations between half the hex alone trials with the other half in fig. 4e, f.

4. Relating contrast enhancement to behavior

The authors propose that the first odor pulse enhances the contrast of the second odor pulse. If so, should the PORs to the target odor then be larger if it is preceded by a distractor odor than when presented alone? The authors should make this comparison. Also, it is interesting that the authors

were able to use digitized PN responses to decode target odor presentation in fig. 6d. For the decoder that predicted PORs in fig. 7, did they use the digital or analog decoder - this is not clearly specified in the text. If the digital decoder was used - this is especially interesting. However, it does raise a question the authors should address: if decoding can be done with binary measures of PN activity, is it accurate to describe the effect of the distractor odor as 'enhancing contrast'. It seems to me that a binary representation is as contrasting as one can get. So is it that the distractor odor enhances contrast, or just makes the target response different? The authors should please clarify their interpretation.

Finally, given that there is only 500 msec separating the two odor pulses, it seems possible that the two odors actually blend into one another, and the effects observed are more about odors mixing with different kinetics, rather than two very separate odors. Are there PID measurements to verify the odor kinetics mean these are really two separate pulses?

Minor comments:

1. Early in the results section, the authors repeatedly use the word 'variable' to describe the alterations in the target representations introduced by different distractors. This is confusing as 'variability' often refers to differences across identical trials, which is not what the authors mean. This passage is particularly problematic: "...responses of individual PNs during [solitary odor presentation] were reliable across trials, they became variable when the same stimulus was delivered with different stimulus histories.'

I suggest using the phrase altered target response in place of variable, or at the very least, rephrasing somehow to be clear that variability here does not refer to trial-to-trial variability.

2. Most of the experiments described in this manuscript and some of the observations have been seen in a very similar study (Broome, ... Laurent, 2006). This should be adequately cited.

3. Fig 2d pink bars are labeled 'hex after...' but should be 'ger after...'

4. In pg. 5, para 2, the authors write 'In some cases, POR response to the distractor diminished the response magnitude to the conditioned stimulus (Fig. 5c; iaa-hex)'. It appears from the panels in figure 5 that all the distractors reduce the POR responses to the target. Either the authors should justify their statement with statistical tests or make a less odor-specific claim.

5. There are behavioral responses to two of the distractors in figure 5. The authors mention this in the section on behavior (pg. 5, para 2) but do not attempt to explain it. For the reader, it might be helpful to mention that this might be a form of generalization and that they later attempt to explain this based on the PN response patterns in fig. 7. Otherwise the reader is left wondering what to think of this until much later in the text.

6. In figs. 6b,d and 7a the left panel is confusingly like a raster plot. Also, they don't add to the story as the authors don't really comment on trial to trial variability either for PN activity or decoder accuracy. Suggest keeping only the panels on the right or more clearly labelling red/blue as decoder predictions.

7. Legend for Sup Fig 4 is incorrectly titled Sup Fig 5 (i.e. 5 appears twice)

Reviewer #3 (Remarks to the Author):

The authors examine how the context of an odor signal may affect the discrimination capability of the insect olfactory periphery (antennal lobe). To do so they stimulate the olfactory periphery in locusts with two types of stimuli: (i) a 5s puff of a target odorant (hexanol or geraniol), or (ii) a 5 s puff of a "distractor" odorant (e.g. apple) followed by a 5s puff of the target odorant. They record the projection neurons and compare the responses to stimulus (i) and (ii). They see differences in the response to the target odorant that depend on the presence or not of the distractor odorant and propose that such differences enhance contrast and reflect a flexible odor code in which the subset of

PN that respond is allowed to change within a given subset unique for each odor.

The data in the first part of the paper is interesting and of interest to the olfaction community. This part however, requires more control and the analysis in general needs to take into account the variation across trials and the possibility that the distractor odor may still be present when the target odorant arrives. In the second part of the paper the authors propose a possible algorithm for coding and decoding odorant. This part is more speculative but should also be of interest to the neuroscience community.

1) Stimulus dynamics. Currently the paper assumes there is no complex stimulus dynamics, i.e. when the puff of target odor arrives the concentration of the distractor odor has returned to zero. Unfortunately, this assumption is not supported by data: no measurement of the time-dependent dynamics of the odorant concentration is provided. This makes it difficult to know to what extent differences in the response to the target odor are due solely to neural response. Even though the authors wait 500 ms between the distractor and the target odor, it is known that many odorants can linger much longer due to interactions between odorant molecules and the various surfaces of the odorant delivery system. If the distractor stimulus is still somewhat present as the target stimulus is starting to arrive then changes seen in the response to the target odor would be due to both the neural dynamics and complex stimulus dynamics. A fast photoionization detector could be used to measure the time dependent concentration and characterize the decay of the odorants at the end of the 1st puff.

2) PN responses vary depending on stimulus history, paragraph 2 / Fig 1c,1d. Another control experiment would be desirable: the distractor stimuli alone, but not followed by the hex target. In some cases, it appears that there is a latency during the 500 ms ISI, but in other cases, the firing persists in the ISI. The conclusion would be more sound if the reader could see what the firing looks like for a few seconds after the removal of distinct distractors, to ensure that the change in firing during hex or ger is not contaminated by the simple act of distractor removal. This would also give an idea of how long it takes for the neuron to reach baseline after the removal of certain distractors. Perhaps the change in firing (suppression or enhancement) during hex / ger is correlated to decay time following distractor removal.

3) Results, PN responses vary depending on stimulus history, paragraph 3 / Fig 1e,1f. It is difficult to believe that PCA is conclusive here, seeing that the top three eigenvectors together account for less than 35% of the variance in the data (compared to 1g-1h, where the results are more sound). Further, though the trajectories are unique, they do not appear particularly well-separated. It would not be surprising to find distinct trajectories, much like these, for different trials of the same history. So it would be helpful to also compare trajectories within distinct trials (not averaged) of the same history, to set a baseline for how much deviation we should expect from the distinct histories. As is, the comparison is not conclusive. If this comparison is sound, there are still some confusions in the figures. What is the timescale of looping – i.e. does it take full 4 s to loop once or does this occur many times in the odor window? Are the large-radius sections of the trajectories the transient portion? Where is the maximum response (first 1 s or so) compared to the adapted response, and where do the odor presentations begin and end? Indeed, the adapted responses might be fairly proximal in this projection, putting the main conclusions into question.

4) Fig 1gh. These figures show nicely how the mean (averaged across trials) response to the target deviates when a distractor is presented before the target odor. However, in order to interpret the scale of these deviations it is necessary to compare them to the difference observed across trials for the same stimulus. Could the authors generate similar plots but comparing responses to the same stimulus across trials?

5) Supp Figure 1ef: The classification analysis and its key result should be explained briefly in the main text since the authors say the classification analysis is quantitative whereas Figure 1ef is qualitative. In particular (related to the previous point) it should be clear to the reader to what extent single trial can be classified. Now it seems only the classification of the means across trials are considered.

6) Stimulus-history dependent contrast enhancement / Fig 2a, b Sort of the same concerns here as 1e-f. Are these variations meaningful? If one plots distinct presentations of the same history, how distinct are the trajectories? This would give a good baseline for comparison. Also, again, the eigenvalues are not that different in a number of these plots, raising the viability of PCA in the first place.

7) Fig 2 Correlation analysis. This analysis focuses on the beginning of the response. This should be justified/motivated in the main text and a similar analysis should be carried for the last second of the response. The differences between early and late response should be analyzed.

8) Temporal response features also vary with stimulus history / Fig 4e-f As a baseline, would be informative to see the correlation between distinct presentations of a unique history. The spike trains certainly suggest that these correlations are higher than the sequential case, but a plot here would give a quantitative comparison.

Minor points:

9) Figures 1ef and similar figures throughout the paper: In general it is difficult for the reader to differentiate the trajectory from the straight lines that join points on the trajectory to the start position. Maybe the trajectories should be plotted using thicker lines and the straight lines should be deemphasized (greyed or made somewhat transparent, or thinner...)

10) fig 2a, first plot, one of the axis labels is missing.

11) Supp Fig 2a: vertical axis should have same scale for all plots to enable direct comparison

Summary of changes:

New data added to the manuscript:

	Type of Change	Previous Submission	Current Submission
Olfactory stimulus characterization	Photoionization detector experiment.	Was not done before.	Responses for solitary presentation of hex, hex after 2oct (delays of 0.5 s, 2 s, 10 s), and hex after iaa (delays of 0.5 s, 2 s, 10 s) were recorded using a fast PID.
Electrophysiology	Projection Neuron (PN) responses to distractor-target odor sequences with longer delay (2s and 10s) between the two stimuli.	85 PNs for ten (distractor, target) odor pairs with 0.5 s delay between distractor odor and target odor.	Additional 104 PNs for 3 (distractor, target) odor pairs with 2 s and 10 s delays between distractor odor and target odor.
Behavior	POR responses to additional distractor-target odor sequences to show response selectivity.	5 distractor odors (2oct, iaa, bzald, cit, and ger) tested for hex trained locusts ($n = 27$).	Additional 5 distractor odors were tested (neem, chex, ger, L-carv, and m-sal) for hex trained locusts ($n = 20$).

1. We characterized the stimulus dynamics when distractor and target odors were presented in a non-overlapping sequence using a fast photoionization detector (PID) as suggested by Reviewers 2 and 3.
2. We have performed additional extracellular recordings from 104 PNs to characterize contrast enhancement for longer delays (2 s and 10 s) between distractor and target odors. After analyzing these new datasets, we found that the contrast enhancement can still be observed with longer delays. These data further support our contrast enhancement results.
3. We have performed new behavioral experiments to test palp opening responses (POR) specificity to the trained odor (hex). These additional experiments, using five additional distractor-target odor sequences, show that conditioned locust PORs are indeed specific to the trained odorant. This was recommended by Reviewer 1.

Reviewers' comments:

Reviewer #1 (Remarks to the Author):

General Comments:

Understanding the neural coding of sensory stimuli is a key aspect in understanding how the nervous system functions. In the field of olfaction, although there have been many studies investigating the neural representation of odors across neural population and time, it is not clear how stimulus identity may be encoded across contexts, while flexibility is allowed to represent the richness of the contexts. In this manuscript, the authors attempted to address this question by investigating the population coding of target odorants among 85 locust antennal lobe olfactory projection neurons (PNs), preceded with 5 distractor odorants from different functional groups with a 500 ms interval.

They found that the PN responses to the same target odorant varied depending on the preceding distractor odorant. Moreover, the presence of the preceding distractor odorants made the PN responses to the target odorant more distinct from the distractor odorants presented alone (contrast enhancement). Intriguingly, they found only about half of the PNs responding to the solitary target odorants consistently responded across stimulus history and they did not uniquely respond to the target odorants. The temporal evolution of the PN responses to the target odorants were also highly variable across different stimulus histories.

Given this divergence of PN encoding of the target odorant depending on distractor identity, would the animal recognize it as the same odorant. The authors performed palp conditioning experiments to investigate, followed by modeling exercises to suggest that a flexible OR of ANDs decoder was able to predict the behavioral responses from the PN neural data.

The study described in this manuscript was significant, experimental design was novel, experimental data was extensive, and statistical analysis and modeling approaches were sound. It also raises interesting questions for future exploration. Please see below for major and minor issues.

We thank the reviewer for the positive evaluation of our work.

Specific Issues:

Major:

1) Palp opening response is extremely reduced as a measure for odor identity. How specific is the POR to hexanol (Figure 5)? How often do the palps open for odorants other than the CS+? For example, in Figure 5 panels b and c, as mentioned in the text, the maxillary palps were already open in response to the distractor odorants. It is not convincingly shown that the locusts actually recognized the target odorant, especially in panel c. Can the authors test more odorants to demonstrate the POR is specific and reliable? This needs to be discussed further.

It would be worth noting that responses of a trained animal to the conditioned stimulus (salivating to an auditory cue or extending proboscis/opening palps in insects) is widely regarded as a measure for stimulus recognition. It is also well-established that there is cross-learning or generalization across stimuli that are considered 'similar'^{1,2}. In our previous work, we had shown that there is cross-learning between odorants in the behavioral POR assay. Notably, which non-trained odorants elicit POR could be predicted based on the overlap in the ensemble PN activity between the untrained and trained odorants³. For example, both 2oct and iaa activate several PNs which also respond to hex. This resulted in false positives in the classification analyses results shown in **Fig. 6d** and **Fig. 7b** (both ON and ON-OFF models).

Note that in the POR responses shown in **Fig. 5b, c**, locusts keep their palps open even after the distractor was terminated. This result indicates that while they generalized their responses to 2oct and iaa, they do respond to the target hexanol pulse as well (else the POR would have terminated after the first stimulus pulse as observed in solitary hex pulse elicited PORs).

In addition, to address this reviewer's concern, we performed a new set of behavioral experiments to further test the specificity of POR to hexanol. In these experiments, we again evaluated locust PORs to hexanol delivered solitarily and in a sequential fashion. Here, we used an additional set of five new distractor odorants: neem oil, cyclohexanone (chex), geraniol (ger), L-carvone (L-carv), and methyl salicylate (m-sal). Training and testing procedures were the same as before. Locust PORs (mean \pm SEM) are shown in **Rebuttal Fig. 1**. **As can be noted, trained locusts only responded to hexanol (the conditioned stimulus) in these new set of experiments.** These new results along with those shown in the main manuscript clearly demonstrate that the locust PORs are indeed specific and reliable, and that the locusts can identify the reinforced odorant consistently.

Rebuttal Figure 1: Locusts were trained to associate hexanol with a food reward using the same protocol as previously described. The distance between the palps was tracked and plotted as a function of time. PORs of trained locusts (mean \pm SEM) to solitary conditioned stimulus presentation (**panel a**) and various distractor-target odor sequences (**panels b-f**) are shown ($n = 20$ locusts). Note that the lag between the two stimuli delivered in sequence is 500 ms (same protocol as before).

2) The modeling results matched the data well, but may or may not be the actual algorithm employed by the brain. Here the modeling is a proposal, not a fact. Therefore, the final sentence in the abstract should be a suggestion rather than a declaration.

Agreed! The point raised by the reviewer is again well taken. We did not mean to imply that the locust brain was following any particular algorithm/decoder. All we wanted to show was that the information was there and a very simple decoder was sufficient to recognize a target odorant encountered in different distractor-target odor sequences. The only caveat we added was that, for any such decoding, activity from a flexible combination of neurons should be taken into account. We modified the last sentence in the abstract as follows:

“In sum, our results suggest that a trade-off between stability and flexibility in sensory coding could be achieved using a simple computational logic.”

Minor (presentation, typo):

1) Figure 1e, f: need to streamline the trajectories. Very difficult to distinguish them with all the spokes and 6 different shades. Supplementary videos or MATLAB figures might help.

We have made the spokes much thinner to emphasize the time evolution. We also made videos of these trajectory plots to help allow observing the same from different angles (**Supplementary Videos 1, 2**).

2) Figure 1g, h: Make the spheres less opaque so the front ones do not completely block the back ones. Again, supplementary MATLAB figures would help so readers can view the plots from varied angles.

We tried making the spheres less opaque, but it did not seem to make much difference. Therefore, we instead made videos to show the figure from different viewing angles (**Supplementary Videos 3, 4**).

3) Supplementary Figure 1a-d: do not address the response rule as described in the second paragraph of Results on page 2.

Agreed! We have replaced these plots with newer analyses and plots that show this response cross-talk clearly.

We plotted the response of each projection neuron to solitary presentations of the distractor odorant along the x-axis (**Rebuttal Fig. 2**). We plotted the PN response to solitary (circle symbols) and sequential (diamonds) presentations of the same target odorant along the y-axis. The symbols associated with each PN's response ([distractor, target alone] and [distractor, target after distractor]) were linked together with a color-coded line. Red lines indicate that PN responses decreased when the same target odorant was presented in a sequential fashion, whereas blue lines indicate the opposite trend. Note that a stronger response to the distractor odorant (higher x-values) resulted in a decreased response to the target odorant which followed (red lines). The opposite trend is typically true for PNs that responded weakly to the distractor stimulus. Therefore, more blue lines are observable for lower x-values (on the left) and more red lines for higher x-values (on the right). Also, note that this trend is stronger when there was a high history-dependent contrast enhancement (shown in **Fig. 2c, d**) in odor-evoked responses to the target odorant (both hex and ger presented after 2oct, iaa and cit distractors)

Rebuttal Figure 2: (a) Response to target vs. response to distractor is plotted for five distractor-target pairs (hex is target odorant in all cases). Values plotted are the maximum spike rate (binned 50 ms) in the first 1 s of response for excitatory neurons to hex ($n = 36$ excitatory PNs to solitary hex). Responses to target alone are plotted as 'circles' and responses to target after distractor are plotted as 'diamonds'. Red lines connecting circles and diamonds indicate that the

response to target after distractor is less than the response to target alone. Blue lines indicate increase in the response for target after distractor when compared to target alone. A very small uniform random noise has been added to jitter the points with same x-values and reduce overlap between colored lines.

(b) Similar plots as **panel a** but plotted when the target odor was ger.

4) Figure 5 (a) legend on page 11: second line missing “to” after “subsequently responded”

Done!

5) Supplementary Figure 4 is mistakenly titled “...5”

Done!

Reviewer #2 (Remarks to the Author):

In this manuscript, the authors describe recordings from the second layer of neurons (PNs) in the locust olfactory system, while delivering single or pairs of odor stimulus pulses. They claim that the PN representation of the second odor pulse (the target) was altered by the specific odor delivered as the first pulse (the distractor) so as to increase the contrast between target and distractor. The authors then try to explain the changed response to the target. These are the most interesting parts of the manuscript to me. The authors then use conditioned responses as a behavioral readout of odor identification and are able to predict performance from alterations in PN responses to the target odor due to the distractor. Overall, the explanation of the history-dependent changes in PN responses to the target odor represents an important addition to the field, but the manuscript requires a few important revisions.

We thank the reviewer for finding parts of the manuscript interesting. We expected some changes in PN responses with stimulus history, but not to the extent observed. But this observation also raised important questions regarding how information related to odor identity is encoded, and on robustness of behavioral recognition. This is what we examined in the later part of the study. We believe that the second part of the study is important as well.

Major comments:

1. Explaining the changed responses to the target

The most interesting part of the manuscript is the attempt to explain the changes in responses to the target odor based on two general observations.

General rule of target response being inversely proportional to distractor response

o This phenomenon should be properly quantified. Please plot new versions of Supp. fig. 1 a, b in main figure to make this point as follows: spike counts (target after distractor – target alone) v/s (distractor - baseline) to show the general rule

o The observation seems anecdotally true, but here are some examples where it appears not to be. Hence, please quantify thoroughly.

In fig. 1 c, d

§ hex after app in PN2

§ ger after cit and ger after app in PN3

§ ger after cit and ger after app in PN4

This issue was also raised by Reviewer 1. We performed a new analysis (**Rebuttal Fig. 2; refer page no. 5**) to show how PN responses change as a function of the distractor response. We believe that this new plot clearly shows the general rule. We thank both reviewers for raising this important issue.

- Contrast enhancement (fig. 2, fig. 4e, f).

The authors should try to use the responses to target alone and distractor alone to predict the response to target after distractor.

For example, a reflection of the distractor representation in N-d space, could be calculated as target after distractor prediction = target vector + (target vector - distractor vector). How well does such a simple estimation match the observed responses to target after distractor?

We performed this analysis requested by the reviewer. We combined the target response vector in each time bin with the (target vector – distractor vector) in that same time bin to generate a target after distractor prediction (**Rebuttal Fig. 3**). As can be noted, in most cases, there is a good match between the estimated and the actual PN responses. The estimated vectors were longer as they were obtained through summation of two vectors.

Rebuttal Figure 3: (a, b) Similar plots as in Fig. 2a, b. Predicted response trajectory is shown in each panel along with trajectories for responses to distractor, target, and target after distractor. The predictions were made for each target after distractor by taking a sum of target vector and (target vector - distractor vector) at each time bin.

Also, if the changes do indeed reflect contrast enhancement, the activity pattern evoked by target after distractor should be similar to target alone but more different from distractor alone. The authors should show that the correlation of target after distractor with target alone remains higher than with distractor.

If the authors are able to concretely and quantitatively show the basis of the changed responses to the target odor due to presentation of the distractor, it would really strengthen the conclusions of the paper.

Fair point. We added the correlation between the target and target after distractor responses (**Rebuttal Fig. 4**). The results exactly matched the expectation of this reviewer: the activity pattern evoked by target after distractor was similar to target alone but more different from distractor alone.

Rebuttal Figure 4: (a) The mean of correlation values between the ensemble PN responses ($n = 85$ PNs) evoked by hexanol and the five distractor cues are shown as bar plots. Error bars indicate \pm s.d across ten trials. The mean odor-evoked responses during the initial 1 s after stimulus introduction were used for computing these correlations. Maroon bars quantify similarities between solitary and sequential presentations of hexanol. Orange bars indicate similarities between the solitary distractor cue and sequential hexanol presentations.

(b) Similar plots as in **panel a** but geraniol is used as the target odorant.

2. Reduce the number of decoders used:

The manuscript uses five different decoders on the PN data collected. The authors should try to reduce the number and complexity of the decoders used, keeping in mind the useful biological property illustrated by each one. The decoders used in fig. 6 and supp. fig. 4d for example, are both trying to make the point that the PN recordings contain enough information to identify the target odor, even after the changes caused by the distractor. One should suffice.

I feel the kernel-based decoder described in supp. Fig. 4 could be dropped or shifted mostly

(text) to the supplement. This would be less distracting for the reader, especially since the text refers only to supp. figures.

We understand the concern of the reviewer. However, the main point we wanted to convey with the four decoding schemes are quite different. The underlying logic in the four classifiers used is as follows:

1. The SVM classifier: This is the best linear classifier. There are eighty-six parameters corresponding to the weights for each neuron, plus a bias term. As the reviewer indicated, this classifier was used to show that the information was there for decoding but the behavioral prediction mismatched with actual behavioral responses observed. The false positives that we observed in the behavioral responses were not predicted using this approach.
2. Flexible decoding scheme (ON response only): Having determined that robust odor recognition was indeed possible, we next sought to determine the simplest possible decoding approach that would generate predictions consistent with the observed POR responses. For this purpose, we used a flexible decoding scheme (with ON responses only; **Fig. 6d**) that used only one tunable parameter. This simple scheme generated predictions that better matched with the actual behavioral responses including the false positives. The only mismatch observed was the response to app. Although this scheme predicted a POR response to app, none were observed in the actual experimental data.
3. Flexible decoding scheme (ON-OFF scheme): In a previous work²⁵, we showed that the ensemble activity during (ON response) and immediately after (OFF response) odor exposure were orthogonal and highly non-overlapping in nature. Notably, we showed that the ON responses were a better indicator of when the locusts opened their palps, and the OFF responses were a better indicator of when the locusts closed their palps. So, we combined the outputs from two flexible decoding classifiers (i.e. flexible decoding with ON and OFF responses; uses two free parameters; **Fig. 7b**). One used the pattern match with the ON-response template of the conditioned stimulus, the other did the same with the OFF-responses elicited by the conditioned stimulus. Notably, for the app odorant, the output of the flexible ON decoder and the flexible OFF decoder canceled each other out, thereby generating more accurate predictions of PORs elicited in our behavioral assay.
4. Digital flexible decoding scheme (ON-OFF scheme): This is the same as classifier 3, but instead of analog spike counts, we used a digital representation for each PN.

'1' -> PN fired about baseline levels (mean +6.5 s.d. of spontaneous activity)

'0' if not.

This was done for each time bin. The weight vectors were same as in the analog version. If m -out-of- n neurons fired synchronously, then that indicated whether the target odorant was present or not (m and n values are listed in the figure). Notably, we found that this digital version of the flexible decoding classifier also generated robust behavioral predictions.

The one in **Supplementary Fig. 4d (previous submission)** is superfluous and has been removed. We have added this explanation to the discussion section.

Also, some clarifications are needed about some of the decoders

- The thresholded OR-OF-AND classifiers described in figs. 6 and 7: How do the POR prediction traces look for the on and off decoders individually? How different are the on and off weight vectors? What fractions of the PNs have weights of 1 as opposed to 0 for each solitary odor?

The POR prediction using the ON responses alone (**Rebuttal Fig. 5a**) will correlate with the classification probability shown in **Fig. 6d**. The POR prediction trace resulted in misclassification for three of the distractors: 2oct, iaa and app. Note that POR for ‘app’ odorant was predicted by this model but not observed in the actual behavioral responses (**Fig. 5f**). Also, note that once the POR starts in this model, it does not return back to baseline.

In our model, the OFF response classification predicts closing of the palps. Hence, without the ON model, this would result in a negative POR. Roughly, the POR responses would be negatively correlated (**Rebuttal Fig. 5b**) with the predicted traces (using OFF model) shown in **Fig. 7b**.

Rebuttal Figure 5: (a) POR prediction using only the ON responses with the ON-OFF model (**Fig. 7c**) is shown. Six panels correspond to six different introductions of hex. The true POR (mean \pm SEM; $n = 27$ locusts) was plotted in black and predicted POR was plotted in red.

Correlation coefficient between means of the true POR and predicted POR is shown on each of the panels as ρ **(b)** Similar plots as in **panel a**, but POR prediction was done using the OFF responses.

The weight vectors used for the ON and OFF models are shown in **Fig. 7a**. Note that the weight vectors for hex-ON model and hex-OFF model are highly non-overlapping (or orthogonal). In **Fig. 7a**, **42%** of PNs have weights of 1 in hex-ON model, while **39%** of PNs have weights of 1 in hex-OFF model.

- Eqn 2 in the methods section does not adequately explain how the weight vector is decided for the SVM decoder. Please explain.

For the plots shown in **Fig. 6a, b**, the two classes we considered were hex ON responses vs. rest. The separating hyperplane was found using an SVM classifier that maximizes the functional margin between the hyperplane and the classes⁴.

$$\gamma_i = y_i(v_{svm}^T x_i + b)$$

Where γ_i is the functional margin for the data point x_i [85-dimensional PN spike counts in a 50 ms time bin], and y_i is the true class label for the data point [1 for hex-ON and -1 for the rest]. The weight vector v_{svm} (also 85 dimensional; i.e. 85 free parameters to tune the model) plus the bias term b (86th model parameter), was calculated by solving the following optimization problem:

$$\min_{y, v_{svm}, b} \frac{1}{2} \|v_{svm}\|^2 + C \sum_{i=1}^m \mu_i$$

$$\text{s.t. } y_i(v_{svm}^T x_i + b) \geq 1 - \mu_i \quad i = 1, \dots, m$$

$$\mu_i \geq 0, i = 1, \dots, m$$

where m is the total number of data points in both the classes and μ_i is the slack variable that help us avoid overfitting while solving this optimization problem. This was done using `fitcsvm` in MATLAB by taking the box constraint to be 0.01. After determining the optimal weight vector v_{svm}^T , we classified response patterns generated during sequential stimulus presentations as follows:

$$h(x(t)) = \text{sign}(v_{svm}^T x(t) + b)$$

where $x(t)$ is the 85-dimensional PN spike counts in a 50 ms time bin, the sign function results in '1' when the input is classified as 'hexanol present' and '0' corresponded to 'hexanol absent.'

- In general, the authors should be more clear about what measure of neuronal response is used as the input to each classifier. In some cases, trajectories are used, but in others, just response amplitude values (I believe - please specify more clearly). It is somewhat jarring for the reader to switch between analyses without more elaboration. For instance, how was the single response amplitude value calculated for use in the LDA plots in fig. 1? The LDA based trial classifier needs some clarification. What was the shape of the matrix on which the LDA was performed? How exactly were the 85 dimensional PN response time-series trial data binned and concatenated in order to perform the LDA? Also, how were the confusion matrices in supp. fig. 1 computed? Were distances to centroid calculated in 85 dimensional LDA component space or just the first three? They should be calculated in the full, 85-d space. I ask this in relation to the fact that all the on-diagonal classification scores are quite low and in fact, not different from background in the case of hex or ger alone or hex or ger after app. In fig. 1 gh, the LDA plots have single points as opposed to the trajectories in fig. 1 ef. How are these single points calculated? Are they trajectories or is there one point for each trial?

Clarification regarding all other analyses:

For all analyses, we counted PN spikes in non-overlapping 50 ms time bins, and averaged them across ten trials. Spike counts of each PN was regarded as a vector component and all PNs recorded were used to generate high-dimensional PN spike count vectors.

The PCA trajectories shown in **Fig. 1e, f** and **Fig. 2a, b** were only used for qualitative understanding of how ensemble PN responses varied with stimulus history. They were not used for any further analyses.

For all correlation and classification analyses (**Fig. 2, 6, 7**), the raw 85-dimensional PN spike count vectors were used.

Clarification regarding LDA plots:

Shape of data matrix:

85 rows (neurons) x 480 columns (6 different presentations of hex/ger and 80 time bins for each odorant (4 s odor puff duration)). Note that the responses over trials were averaged for each odorant.

This resulted in a 6-class discrimination problem: one solitary hex/ger presentation and 5 sequential presentation of the same target odorant. So, the LDA dimensionality reduction can result in at most a five-dimensional approximation of the original dataset (rank = number of classes – 1). Projections along the top three eigenvectors of the ($S_w^{-1}S_B$) matrix were chosen for plots shown in **Fig. 1g, h**. Note that S_w is the within-class scatter matrix and S_B is the between-class scatter matrix⁴.

In the previous manuscript version, the confusion matrix was generated using the LDA projected data points in 3D. However, **Rebuttal Fig. 6** shows the confusion matrices when the distance from the centroid was computed using all the 85 dimensions (i.e. no LDA dimensionality reduction). It is clear that the confusion matrices are more diagonal in this case, which suggests

that hex (and ger) presented with six different stimulus histories tended to form six distinct clusters in the 85-d space. We thank the reviewer for this suggestion. These details have been added to the methods section of the manuscript. The confusion matrix shown in **Supplementary Fig. 1** have now been updated to show results from the 85-dimensional space.

To be consistent everywhere, we have now used dimensionality reduction methods (PCA and LDA) to qualitatively understand data structure, but performed ALL quantification (i.e. correlation and classification analyses) using the 85 dimensional PN response vectors.

Rebuttal Figure 6: (a) The separability of the PN responses (six categories: hex(solitary) and five sequential conditions of hex) is quantified and shown as a confusion matrix. Rows correspond to the actual stimulus identity and the columns indicate the predicted stimulus identity. A nearest centroid method with leave-one-out cross-validation in 85-D space was used for generating these classification results. Note that the confusion matrix is mostly diagonal indicating that the PN responses evoked by the same stimulus presented with different histories are distinct.

(d) Similar plot as in **panel a** but the confusion matrix analyzing the response separability of the solitary and sequential geraniol presentations is shown.

3. Recommend tighter focus on a central message in the manuscript
 fig. 3 does not add value to the manuscript. Combinatorial coding in PNs is well established. The lack of unique, consistent responders is unsurprising. This is especially the case given that the authors tested a limited, arbitrarily chosen number of distractor odors. It can be argued that given enough distractor odors, it would be impossible to find a unique responder. I recommend moving this section to the supplement.

It is well established that most odorants activate combinations of PNs. While the result that there is a lack of unique, consistent responders is logical extrapolation of results shown by the contrast enhancement result reported in this manuscript, to our knowledge, this has not been shown. We will use the following quote from Reviewer 1's comments to strengthen our case here:

“Intriguingly, they found only about half of the PNs responding to the solitary target odorants consistently responded across stimulus history and they did not uniquely respond to the target odorants.”

Requoting this reviewer: “It can be argued that given enough distractor odors, it would be impossible to find a unique responder.”

This is exactly the point that we are trying to make - that there should be flexibility in the decoding scheme as individual PN responses are unstable across conditions (the second part of the manuscript). We are glad the reviewer agrees with our interpretation. However, this expectation follows from the results shown in this manuscript, and does not constitute a well-established result/understanding in this field. Therefore, we would still like to keep these results in the main manuscript.

The section on ‘Temporal response features also vary with stimulus history’ is not sufficiently supported by the analysis in fig. 4 and does not really add to the central theme of the manuscript.

We disagree with the reviewer. The same reasoning as in the previous point also holds true here. The main point is that temporal response properties of individual PNs (their latency and spike train structure) vary across stimulus histories.

To really make the point about changing temporal trajectories, a clearer analysis than the slightly biased correlation distributions would be needed. I suggest removing this section from the main manuscript.

We don’t understand why the reviewer finds the correlation analysis ‘biased.’ Note that we did not perform any selection/pre-screening of PN responses. For each PN, we took the spike train structure over the entire target stimulus presentation window (4 s; 1x80 vector for each PN) and compared them across conditions. Quantifying how much the temporal response properties vary with history is also an important result that we would like to keep in the main manuscript.

If the authors want to make this point, they could consider quantifying deviations of the mean target after distractor trajectory from the mean target alone trajectory and comparing those with variability in target alone trajectories. Also consider overlaying a trace for the distribution of correlations between half the hex alone trials with the other half in fig. 4e, f.

Note that the analyses in **Fig. 4** were all done to characterize variability of two different temporal response features (latency and spike patterns over time) **at the individual PN level**.

We have done other analyses to examine how PN ensemble response vary across trials and compare the same with odor-evoked response variations across stimulus histories. We plotted the distributions of inter-trial PN spike train correlations along with the inter-condition PN response correlations (**Rebuttal Fig. 7**). We observed that at the individual PN level, trial-to-trial correlation distributions were only modestly better than inter-condition correlation distributions.

However, we found that the distributions of inter-trial and inter-condition are statistically distinguishable in six (out of ten) cases (two-sample Kolmogorov–Smirnov test, $P < 0.05$, $n = 85$).

Rebuttal Figure 7: (a) Distributions of correlation values between inter-condition (colored) and inter-trial (gray) PN spike trails are shown. This analysis is similar to the one done for generating Fig. 4e, f. For computing correlation between trials, for each PN, we compared the similarity between the mean PN responses in the first five trials (trials 1-5) with the mean response in the remaining five trials (trials 6-10) of hex(solitary) stimulus. Similarly, the correlations for inter-condition correlations (plotted in orange) were calculated by estimating the similarity between the mean PN response in the first five trials (trials 1-5) of hex(solitary) exposures with the mean response of the last five trials (trials 6-10) of sequential hex presentations. This was again done for each PN and for each sequential hex presentation to generate the five orange distributions shown in the plot.

(b) Similar plots as in panel a but analyzing PN responses to geraniol.

Table 1: Comparison of PN response correlation distributions:

	Mean inter-trial correlation (only for target odorant)	Mean inter-condition correlation	p-value	KS-statistic
hex(2oct)	0.3066	0.0937	1.3 e-06	0.4
hex(iaa)	0.3066	0.1892	0.0090	0.2471
hex(bzald)	0.3066	0.1662	0.0053	0.2588
hex(cit)	0.3066	0.1673	0.0090	0.2471
hex(app)	0.3066	0.2431	0.2512	0.1529
ger(2oct)	0.1527	0.0444	0.0239	0.2235
ger(iaa)	0.1527	0.1094	0.7033	0.1059
ger(bzald)	0.1527	0.0967	0.5719	0.1176
ger(cit)	0.1527	0.0770	0.0148	0.2353
ger(app)	0.1527	0.1212	0.8274	0.0941

We also compared the similarity between ensemble PN neural activities observed in different time bins (**Rebuttal Fig. 8a**). First, we computed the correlation between trial averaged responses in different time bins observed during the 4 s of hexanol presentation (solitary condition). As can be noted, the initial ~500 ms were less correlated with the responses observed during the later epochs. However, the responses evoked during the later temporal epochs were well correlated amongst themselves. This overall correlation structure, appears to be conserved across sequential introductions of the hexanol with different distractor stimuli. While the correlation with solitary hexanol responses were uniformly high, some conditions had less pairwise-similarity (e.g. hex after iaa vs hex after cit, hex after 2oct and hex after bzald). On the other hand, similarity between ensemble neural activities elicited by different odorants remained relatively low across all time bins (**Rebuttal Fig. 8b**).

Rebuttal Figure 8: (a) Correlations between ensemble PN response vectors evoked by an odorant is shown for each time bin in 4 s duration (80 time bins). Spike counts averaged across trials ($n = 10$ trials) were used for this analysis. Self-correlation and cross-correlations were computed among all the six presentations of hex.

(b) Similar plots as in **panel a** but the correlations between hexanol and the five distractor odorants are shown.

Lastly, we performed the same analysis we did in **Fig. 4**. We identified early and late responders across different sets of trials (**Rebuttal Fig. 9**). As can be noted, the set of early responders remained highly consistent across different blocks of trials (**Rebuttal Figure 9** vs. **Fig. 4a**).

Rebuttal Figure 9: Trial-averaged PN responses are shown (similar to **Fig. 4a,b**) for two sets of trials (trials 1-5 and trials 6-10). Bar-code on the right of each panel shows the early responsive PNs (response latency ≤ 600 ms) in white and late/non-response PNs shown in black.

4. Relating contrast enhancement to behavior

The authors propose that the first odor pulse enhances the contrast of the second odor pulse. If so, should the PORs to the target odor then be larger if it is preceded by a distractor odor than when presented alone? The authors should make this comparison.

Our result revealed that the neural responses evoked by an odorant changed substantially when the same odorant was presented with different stimulus histories. Given this alteration in odor-evoked responses, we were not sure whether a trained locust could recognize and respond with PORs when the same conditioned stimulus was presented with varying histories. Our behavioral experiments were therefore important to show that locusts were indeed capable of performing this recognition task.

We are not sure why the reviewer is expecting a higher POR to the target when it is preceded by a distractor odor. Note that response to hexanol did not increase, only that its response became more unique with respect to the distractor that preceded it!

Also, it is interesting that the authors were able to use digitized PN responses to decode target odor presentation in fig. 6d. For the decoder that predicted PORs in fig. 7, did they use the digital or analog decoder - this is not clearly specified in the text. If the digital decoder was used - this is especially interesting. However, it does raise a question the authors should address: if

decoding can be done with binary measures of PN activity, is it accurate to describe the effect of the distractor odor as 'enhancing contrast'. It seems to me that a binary representation is as contrasting as one can get. So is it that the distractor odor enhances contrast, or just makes the target response different? The authors should please clarify their interpretation.

Results shown in **Fig. 7** were generated using an analog decoder.

Results shown in **Supplementary Fig. 6** were generated using a digital decoder. As can be noted, these two results are qualitatively similar. So, yes, even we were surprised that the digital decoder can indeed generate such robust recognition results.

Binary representation does not imply contrast!

Pattern A: [1 1 1 0 0 0]

Pattern B: [0 1 1 1 1 0]

Contrast enhanced pattern B (with respect to pattern A): [0 0 0 1 1 0];

Note that common channels 2, 3 has been suppressed to enhance the contrast of the binary patterns.

Finally, given that there is only 500 msec separating the two odor pulses, it seems possible that the two odors actually blend into one another, and the effects observed are more about odors mixing with different kinetics, rather than two very separate odors. Are there PID measurements to verify the odor kinetics mean these are really two separate pulses?

This is a valid concern. We looked in to this issue carefully. First, we used photoionization detector (PID) to examine how rapidly the odor pulses decay after closing of the olfactometer valve. As can be observed from **Rebuttal Fig. 10**, for hex and iaa, these returns back to baseline were rapid, whereas 2oct had a longer response decay time constant. We presented distractor-target odor pulses as we did in our electrophysiology experiments. We found that for the 0.5 s lag between the two pulses, there was interaction between the first 2oct pulse and the second hex pulse. This reduced substantially when the lag between the two pulses was increased to 2 s and 10 s. For the iaa-hex sequence, iaa decayed rapidly. So, there was minimal interaction between the two pulses even for 0.5 s lag.

Rebuttal Figure 10: (a, b, c) Photoionization detector (PID) responses are shown. The color bars indicate when a 4 s odor puff was presented. The three traces shown correspond to hex, 2oct and iaa presentations, respectively. Mean across five trials is plotted in each panel. **(d, e, f)** Similar PID traces are shown for 2oct-hex (distractor-target) odor sequences with three different lags between the two stimuli: 0.5 s, 2 s, and 10 s. **(g, h, i)** Similar plots as **panels d, e, f**, but for a different distractor-target (iaa-hex) odor sequence.

Given the slow decay of the 2oct pulse, we examined whether the change observed during sequential presentation was similar to what would be observed for a binary mixture of two odorants. To understand this, we reanalyzed one of our previously published dataset⁵. This dataset had responses to both hexanol (hex), 2octanol (2oct), a binary mixture of hexanol and 2octanol (hex-2oct), and hexanol presented 0.25 s and 0.5 s after the 2oct pulse.

Ensemble PN response to this stimulus set is shown in **Rebuttal Fig. 11**. As can be noted, the binary mixture of hex-2oct was an additive combination of two component odorants and therefore generated a response trajectory in-between the two component trajectory responses (i.e. green trajectory (hex-2oct mixture) is in-between the red (hex) and the blue

(2oct) trajectories). However, note that for both hexanol presentations with 0.25 and 0.5 s lag following 2oct (distractor – 0.25 s – target and distractor – 0.5 s – target), the responses elicited following hex presentations were more similar to hexanol than the binary mixture of the distractor. Further, the responses shifted away from the 2oct response trajectory (i.e. magenta trajectory).

Rebuttal Figure 11: (a, b) Similar plots as in Fig. 2a but showing ensemble response trajectories for the following stimuli: hex (red), 2oct (blue), a binary mixture of hex and 2oct (hex-2oct; green), 2oct – 0.25 s – hex (purple; top panel) and 2oct – 0.5 s – hex (purple; bottom panel).

(c) Correlation values (mean ± s.d.; $n = 10$ trials) for binary mixture response, hex response, hex after 2oct (0.25 s) response, and hex after 2oct (0.5 s) response were calculated with 2oct and are plotted. We have used the initial 1 s of response to compute these correlations. Asterisks indicate a significant decrease in the correlation ($*P < 0.0125$ (Bonferroni corrected for four comparisons), t-tests, $n = 10$ trials).

Finally, we delivered distractor-target sequences with larger lag between the two stimuli (2 s and 10 s). We collected a new dataset to examine this. As can be noted, these results (**Rebuttal Fig. 12a, b**) are very similar to the results shown in the manuscript, and contrast enhancement was observed even for these cases. Quantifications of these results are shown in **Rebuttal Fig. 12c**.

Rebuttal Figure 12: (a, b) Response trajectories generated by two additional sequential presentations of hex are plotted: 2oct – 2 s – hex and 2oct – 10 s – hex. Similar method (as in **Fig. 2a**) was followed to analyze this dataset. **(c)** The mean of correlation values between the ensemble PN responses ($n = 104$ PNs) evoked by hexanol and the five distractor cues are shown as bar plots. Error bars indicate \pm s.d across ten trials. The mean odor-evoked responses during the initial 1 s after stimulus introduction were used for computing these correlations. The three bars shown for each distractor indicate similarity between the distractor and solitary presentation of hex (i.e. target), distractor and sequential presentation of hex after a delay of 2 s, and after a delay of 10 s. The correlations

are shown for three distractors 2oct, iaa, and chex. Asterisks indicate a significant decrease in the correlation ($*P < 0.025$ (Bonferroni corrected for two comparisons), t-tests, $n = 10$ trials).

Minor comments:

1. Early in the results section, the authors repeatedly use the word 'variable' to describe the alterations in the target representations introduced by different distractors. This is confusing as 'variability' often refers to differences across identical trials, which is not what the authors mean. This passage is particularly problematic: "...responses of individual PNs during [solitary odor presentation] were reliable across trials, they became variable when the same stimulus was delivered with different stimulus histories.'

I suggest using the phrase altered target response in place of variable, or at the very least, rephrasing somehow to be clear that variability here does not refer to trial-to-trial variability.

Fixed!

2. Most of the experiments described in this manuscript and some of the observations have been seen in a very similar study (Broome, ... Laurent, 2006). This should be adequately cited.

We have cited this work wherever appropriate. We note that although that study also used stimulus sequences, their results did not reveal the contrast enhancement result, probably because both overlapping and non-overlapping odor sequences were used and analyzed together. As shown in **Rebuttal Fig. 11**, these stimuli are processed quite differently: one as a binary mixture (stimulus 1 + stimulus 2) and another as a contrast enhanced version of the second stimulus in the sequence (stimulus 2+ (stimulus 2- stimulus1)). Note also that there was no behavioral validation done in that study.

3. Fig 2d pink bars are labeled 'hex after...' but should be 'ger after...'

Corrected!

4. In pg. 5, para 2, the authors write 'In some cases, POR response to the distractor diminished the response magnitude to the conditioned stimulus (Fig. 5c; iaa-hex)'. It appears from the panels in figure 5 that all the distractors reduce the POR responses to the target. Either the authors should justify their statement with statistical tests or make a less odor-specific claim.

We only wanted to emphasize the stable recognition of hex in all the cases. Other odor specific claims are not central and were not analyzed for statistical significance and have been removed.

5. There are behavioral responses to two of the distractors in figure 5. The authors mention this in the section on behavior (pg. 5, para 2) but do not attempt to explain it. For the reader, it might be helpful to mention that this might be a form of generalization and that they later attempt to

explain this based on the PN response patterns in fig. 7. Otherwise the reader is left wondering what to think of this until much later in the text.

Fair point! We have fixed this in the revised version.

6. In figs. 6b,d and 7a the left panel is confusingly like a raster plot. Also, they don't add to the story as the authors don't really comment on trial to trial variability either for PN activity or decoder accuracy. Suggest keeping only the panels on the right or more clearly labelling red/blue as decoder predictions.

We have removed the left panels in both **Fig. 6b,d** and **Fig. 7a**

7. Legend for Sup Fig 4 is incorrectly titled Sup Fig 5 (i.e. 5 appears twice)

Corrected!

Reviewer #3 (Remarks to the Author):

The authors examine how the context of an odor signal may affect the discrimination capability of the insect olfactory periphery (antennal lobe). To do so they stimulate the olfactory periphery in locusts with two types of stimuli: (i) a 5s puff of a target odorant (hexanol or geraniol), or (ii) a 5 s puff of a “distractor” odorant (e.g. apple) followed by a 5s puff of the target odorant. They record the projection neurons and compare the responses to stimulus (i) and (ii). They see differences in the response to the target odorant that depend on the presence or not of the distractor odorant and propose that such differences enhance contrast and reflect a flexible odor code in which the subset of PN that respond is allowed to change within a given subset unique for each odor.

The data in the first part of the paper is interesting and of interest to the olfaction community. This part however, requires more control and the analysis in general needs to take into account the variation across trials and the possibility that the distractor odor may still be present when the target odorant arrives. In the second part of the paper the authors propose a possible algorithm for coding and decoding odorant. This part is more speculative but should also be of interest to the neuroscience community.

We thank the reviewer for finding the data interesting and her/his positive evaluation of our work.

1) Stimulus dynamics. Currently the paper assumes there is no complex stimulus dynamics, i.e. when the puff of target odor arrives the concentration of the distractor odor has returned to zero. Unfortunately, this assumption is not supported by data: no measurement of the time-dependent dynamics of the odorant concentration is provided. This makes it difficult to know to what extent differences in the response to the target odor are due solely to neural response. Even though the authors wait 500 ms between the distractor and the target odor, it is known that many odorants can linger much longer due to interactions between odorant molecules and the various surfaces of the odorant delivery system. If the distractor stimulus is still somewhat present as the target stimulus is starting to arrive then changes seen in the response to the target odor would be due to both the neural dynamics and complex stimulus dynamics. A fast photoionization detector could be used to measure the time dependent concentration and characterize the decay of the odorants at the end of the 1st puff.

This is a valid point that was also raised by reviewer 2. We did precisely these experiments and others to show that our interpretation of results is still valid. See **Rebuttal Figs. 10 –12.**

2) PN responses vary depending on stimulus history, paragraph 2 / Fig 1c,1d. Another control experiment would be desirable: the distractor stimuli alone, but not followed by the hex target. In some cases, it appears that there is a latency during the 500 ms ISI, but in other cases, the firing persists in the ISI. The conclusion would be more sound if the reader could see what the firing looks like for a few seconds after the removal of distinct distractors, to ensure that the change in firing during hex or ger is not contaminated by the simple act of distractor removal. This would also give an idea of how long it takes for the neuron to reach baseline after the

removal of certain distractors. Perhaps the change in firing (suppression or enhancement) during hex / ger is correlated to decay time following distractor removal.

We do have this data (**Rebuttal Fig. 13**). In general, most odorants evoke an 'OFF response' following stimulus termination. As we discussed in our previous submission this OFF response of the first stimulus could indeed interfere with how subsequent stimuli are being processed (as alluded by this reviewer). Two take-home points are worth highlighting here:

(i) Even if the contrast enhancement was just due to interaction between the 'ON response' of the current stimulus with the 'OFF response' of the preceding stimulus, this happens for most of the odor sequences presented. Therefore, one computational role for the strong neural firing following stimulus termination could be contrast enhancement of any stimulus that would be encountered subsequently (smelling of coffee bean in the perfume shop is a good analogy here). Also, since the behavioral recognition happens so rapidly, the relevance of the neural activity that occurs well after the stimulus onset (including those that happen following termination) is not clear. So, our results are important even in this scenario.

(ii) We also did additional experiments to understand whether contrast enhancement could happen when the delay or lag between the two stimuli delivered in sequence was increased. As shown in **Rebuttal Fig. 12**, our results intriguingly indicate that contrast enhancement occurs even for such delayed presentation of the target stimulus (2 s and 10 s delay between distractor and target odorants)!

Rebuttal Figure 13: Mean firing rate across all neurons (85 PNs) and trials (10 trials) are shown as a function of time for all odors examined in this study. A 3-point moving average filter was used to smoothen the plots shown here.

3) Results, PN responses vary depending on stimulus history, paragraph 3 / Fig 1e,1f. It is difficult to believe that PCA is conclusive here, seeing that the top three eigenvectors together account for less than 35% of the variance in the data (compared to 1g-1h, where the results are more sound). Further, though the trajectories are unique, they do not appear particularly well-separated. It would not be surprising to find distinct trajectories, much like these, for different trials of the same history. So it would be helpful to also compare trajectories within distinct trials (not averaged) of the same history, to set a baseline for how much deviation we should expect from the distinct histories. As is, the comparison is not conclusive. If this comparison is sound, there are still some confusions in the figures. What is the timescale of looping – i.e. does it take full 4 s to loop once or does this occur many times in the odor window? Are the large-radius sections of the trajectories the transient portion? Where is the maximum response (first 1 s or so) compared to the adapted response, and where do the odor presentations begin and end?

Indeed, the adapted responses might be fairly proximal in this projection, putting the main conclusions into question.

The PCA trajectories were shown to gain qualitative understanding of the data. All quantifications, including results from correlation analysis shown in **Fig. 2c, d** were computed in the 85-dimensional space. These quantitative results, which included the trial-to-trial variability, are in agreement with the PCA trajectory plots!

However, the point raised by the reviewer is an important one that needs more careful investigation. So, we split the ten trials into three sets: trials 1 - 3, trials 4 - 6, and trials 7 - 10. Note that the trajectories shown (**Rebuttal Fig. 14**) for these three sets of trials convey exactly the same interpretations as the trial-averaged PCA trajectories that were shown in the manuscript. We have added these figures to the Supplementary Figures.

Rebuttal Figure 14: (a) Similar trajectory plots as shown in **Fig. 2a, b** but trial-to-trial variations are included. Population PN responses generated for the distractor, hex, and the sequential presentation of hex for three sets of trials are shown: mean of trials 1-3, mean of trials 4-6, and mean of trials 7-10. The darker colored traces correspond to the earlier trial sets.

(b) Similar plots as in **panel a** but shown for five sequential presentations of geraniol.

Rebuttal Figure 15: (a) Comparison of combinatorial PN response profiles activated by the same odorant across trials, same odorant across stimulus histories, and between different odorants is shown as a function of trial number. Note that all comparisons are made with respect to the ensemble PN responses elicited by solitary presentation of hexanol in the very first trial.

(b) Similar plots as **panel a** but plotted when the target odor is ger.

To systematically compare response variability across different trials with those observed across stimulus history, we used a correlation analysis. First, we computed the mean ensemble response for each trial. We then computed the correlation between the ensemble responses observed in any trial with the PN spiking activity in the first trial of the target odorant (hex or ger solitary presentation). We plotted (**Rebuttal Fig. 15**) how correlation changed as a function of trial number, across stimulus history and between different odorants. As can be noted, correlation between PN spiking activity during different trials when the same stimulus was presented remained relatively high. The correlation reduced when the comparison was made across presentations of the same target odorant in different distractor-target odor sequences. Correlations between odorants were relatively low.

In sum, these results clearly show that the variability across conditions were much greater than those observed across trials but was less than those between odorants.

Regarding evolution over time, the trajectories plotted in **Figs. 1 & 2** show the ensemble neural response during the entire 4 s odor presentation window. Yes, the large radius sections of the trajectories correspond to the transient portion of the neural response, which usually lasts for the first 2 s.

Finally, ‘regarding the adapted response being fairly proximal,’ we found that to be not true. The cross-correlation between target after distractor vs. distractor reduced even when the adapted response (i.e. response during the last 1 s of odor presentation) was examined (**Rebuttal Fig. 16**).

Rebuttal Figure 16: (a,b) Similar plots as in **Fig. 2c,d**, but the correlations are now computed using the mean odor-evoked response during the last 1 s of odor exposure window.

4) Fig 1gh. These figures show nicely how the mean (averaged across trials) response to the target deviates when a distractor is presented before the target odor. However, in order to interpret the scale of these deviations it is necessary to compare them to the difference observed across trials for the same stimulus. Could the authors generate similar plots but comparing responses to the same stimulus across trials?

Done! We averaged three sets of trials: set 1 – trials 1 to 3, set 2 – trials 4 to 6, and set 3 – trials 7 to 10. We performed the LDA analysis and the results are shown in **Rebuttal Fig. 17 (Supplementary Videos 5, 6)**. The confusion matrix plotted (**Rebuttal Fig. 6**) exactly quantifies the trial-to-trial variability. This analysis was performed using leave-one-trial-out cross validation method. High values in the diagonal elements indicate that the clusters are separable.

Rebuttal Figure 17: (a) Population PN responses are shown after linear discriminant analysis dimensionality reduction ($n = 85$ PNs) for three sets of trials: set 1 – mean of trials 1-3, set 2 – mean of trials 4-6, set 3 – mean of trials 7-10. Each 3D-shape represents an eighty-five-dimensional PN activity vector in a 50 ms time bin. Eighty data points corresponding to ensemble neural activities evoked during 4 s of hexanol are shown. Different symbols represent different sets of trials, different colors indicate hex response following different stimulus histories.

(b) Similar plot as in **panel a**, but showing responses elicited during geraniol exposures.

5) Supp Figure 1ef: The classification analysis and its key result should be explained briefly in the main text since the authors say the classification analysis is quantitative whereas Figure 1ef is qualitative. In particular (related to the previous point) it should be clear to the reader to what extent single trial can be classified. Now it seems only the classification of the means across trials are considered.

The confusion matrices (shown in **Supplemental Fig. 1**) were computed by using the leave-one-trial-out method. The centroid of the clusters formed from the mean of 9 trials was used to calculate the training templates and the left-out trial was classified. **Rebuttal Fig. 6** shows the confusion matrices when the distance from the centroid was computed using all the 85 dimensions. It is clear from the plot that the confusion matrices are diagonal, which suggests that all the six conditions of hex (and ger) tended to form six separable clusters in the 85-d space.

6) Stimulus-history dependent contrast enhancement / Fig 2a, b Sort of the same concerns here as 1e-f. Are these variations meaningful? If one plots distinct presentations of the same history, how distinct are the trajectories? This would give a good baseline for comparison. Also, again, the eigenvalues are not that different in a number of these plots, raising the viability of PCA in the first place.

See response above to concern #3, 4 (refer **Rebuttal Figures 14 – 16**).

7) Fig 2 Correlation analysis. This analysis focuses on the beginning of the response. This should be justified/motivated in the main text and a similar analysis should be carried for the last second of the response. The differences between early and late response should be analyzed.

Behavioral odor recognition is rapid with a median latency of ~600 ms. Hence, we focused on the early portion of the responses. As suggested, we also performed the correlation analysis using the last 1 s of response (refer **Rebuttal Fig. 16**). The plot shows qualitatively similar results.

8) Temporal response features also vary with stimulus history / Fig 4e-f As a baseline, would be informative to see the correlation between distinct presentations of a unique history. The spike trains certainly suggest that these correlations are higher than the sequential case, but a plot here would give a quantitative comparison.

Similar question was raised by Reviewer #2. We performed new analysis to show the inter-trial variations along with inter-condition variations of temporal patterns (**Rebuttal Fig. 7**). We found that these two distributions are distinguishable in most cases, and the mean correlation of inter-trial distribution is always higher than inter-condition distribution.

Minor points:

9) Figures 1ef and similar figures throughout the paper: In general it is difficult for the reader to differentiate the trajectory from the straight lines that join points on the trajectory to the start position. Maybe the trajectories should be plotted using thicker lines and the straight lines should be deemphasized (greyed or made somewhat transparent, or thinner...)

We have made the spokes much thinner to emphasize the time evolution.

10) fig 2a, first plot, one of the axis labels is missing.

Fixed!

11) Supp Fig 2a: vertical axis should have same scale for all plots to enable direct comparison

We changed the plots in **Supplemental Fig. 3a, b** (note that the supplementary figure numbering has been shifted after revision) to have the same range along the y-axes.

REFERENCES:

- 1 Daly, K. C., Chandra, S., Durtschi, M. L. & Smith, B. H. The generalization of an olfactory-based conditioned response reveals unique but overlapping odour representations in the moth *Manduca sexta*. *The Journal of experimental biology* **204**, 3085-3095 (2001).
- 2 Guerrieri, F., Schubert, M., Sandoz, J. C. & Giurfa, M. Perceptual and neural olfactory similarity in honeybees. *PLoS biology* **3**, e60, doi:10.1371/journal.pbio.0030060 (2005).
- 3 Saha, D. *et al.* Behavioural correlates of combinatorial versus temporal features of odour codes. *Nature communications* **6**, 6953, doi:10.1038/ncomms7953 (2015).
- 4 Bishop, C. M. *Pattern recognition and machine learning*. (springer, 2006).
- 5 Saha, D. *et al.* A spatiotemporal coding mechanism for background-invariant odor recognition. *Nature neuroscience* **16**, 1830-1839, doi:10.1038/nn.3570 (2013).

REVIEWERS' COMMENTS:

Reviewer #1 (Remarks to the Author):

All my concerns have been adequately addressed in the rebuttal. The authors also performed substantial additional experiments and analysis. It is recommended that this study is published without further revision.

Reviewer #2 (Remarks to the Author):

NOTE: a formatting-preserved version of the review has been attached as a .pdf file.

I want to note at the beginning that I find the results reported here very interesting and support the publication of this manuscript. However, my goal is for all the claims reported in the manuscript to be rigorously supported by the data, analyzed appropriately. In the first round of revisions, the authors have adequately addressed almost all the concerns/suggestions I had. I thank them for their transparency and diligence. There are two remaining points though, where I still feel some small changes would improve the manuscript. The changes suggested are small.

1. The analysis reported in rebuttal figure 2 (Supplementary figure 1 a, b) still fails to quantify the extent to which the data supports the general rule. The authors' claim is justified if in general, the population of blue lines is to the left of the population of red lines. This has not been systematically quantified and in fact, doesn't seem to be the case for hex-app in panel a and ger-benzald in panel b. It should be straightforward to compute means of deviation from baseline for distractors and deviation from normal responses for targets. Then the authors could correlate these two deviations and make a statistically supported claim. This is currently not the case, but looking at the plots, it seems the result of a statistical analysis should be in line with what the authors are claiming.

2. In rebuttal figure 3, the authors show that post-distractor, target responses are sub-linearly predictable from target alone and distractor alone responses. I find this very interesting and am disappointed that it will not be included even as a supplementary figure. Related to this, rebuttal figure 4, also does not make it to the manuscript.

Main figure 2 shows that target responses after distractor presentation are changed from the target alone response. This alone does not imply contrast enhancement. For functionally useful contrast enhancement, the target odor representation must systematically move away from the distractor representation, while remaining close to the original target representation. The analysis already done by the authors in rebuttal figures 3 and 4 show exactly this! It would be a pity for them to not be included in the publication. I urge the authors to re-consider including these analyses.

As an aside, I noticed that the data in rebuttal figure 3 panel b seems identical to main figure 2, panel d. In case the authors decide to include this figure in the supplement, they should rectify this error.

This point is not about an un-addressed concern, it is a clarification. On page 15 of the rebuttal, the authors' response highlights a point where I failed to be clear in my original review. Referring to main figure 4, panels e and f, the authors argue that the bias of the correlation coefficient distribution towards 0 rather than 1 indicates changed temporal response patterns. This is the bias I was referring to in my original review. I pointed out that merely having correlation coefficients smaller than 1 does not indicate systematic changes in response patterns in time. Differences between repeat neuronal response measurements to the same stimulus can arise due to a variety of noise sources. One way to address this is to compare the distribution of correlation coefficients between repeated target alone

presentations with the distribution of correlation coefficients between target alone and target with distractor trials. The authors have done this analysis in rebuttal figure 7 a and b (supplementary figure 11).

I want to note at the beginning that I find the results reported here very interesting and support the publication of this manuscript. However, my goal is for all the claims reported in the manuscript to be rigorously supported by the data, analyzed appropriately. In the first round of revisions, the authors have adequately addressed almost all the concerns/suggestions I had. I thank them for their transparency and diligence. There are two remaining points though, where I still feel some small changes would improve the manuscript. The changes suggested are small.

1. The analysis reported in rebuttal figure 2 (Supplementary figure 1 a, b) still fails to quantify the extent to which the data supports the general rule. The authors' claim is justified if in general, the population of blue lines is to the left of the population of red lines. This has not been systematically quantified and in fact, doesn't seem to be the case for hex-app in panel a and gerbenzald in panel b. It should be straightforward to compute means of deviation from baseline for distractors and deviation from normal responses for targets. Then the authors could correlate these two deviations and make a statistically supported claim. This is currently not the case, but looking at the plots, it seems the result of a statistical analysis should be in line with what the authors are claiming.
2. In rebuttal figure 3, the authors show that post-distractor, target responses are sub-linearly predictable from target alone and distractor alone responses. I find this very interesting and am disappointed that it will not be included even as a supplementary figure. Related to this, rebuttal figure 4, also does not make it to the manuscript.

Main figure 2 shows that target responses after distractor presentation are changed from the target alone response. This alone does not imply contrast enhancement. For functionally useful contrast enhancement, the target odor representation must systematically move away from the distractor representation, while remaining close to the original target representation. The analysis already done by the authors in rebuttal figures 3 and 4 show exactly this! It would be a pity for them to not be included in the publication. I urge the authors to re-consider including these analyses.

As an aside, I noticed that the data in rebuttal figure 3 panel b seems identical to main figure 2, panel d. In case the authors decide to include this figure in the supplement, they should rectify this error.

This point is not about an un-addressed concern, it is a clarification. On page 15 of the rebuttal, the authors' response highlights a point where I failed to be clear in my original review. Referring to main figure 4, panels e and f, the authors argue that the bias of the correlation coefficient distribution towards 0 rather than 1 indicates changed temporal response patterns. This is the bias I was referring to in my original review. I pointed out that merely having correlation coefficients smaller than 1 does not indicate systematic changes in response patterns in time. Differences between repeat neuronal response measurements to the same stimulus can arise due to a variety of noise sources. One way to address this is to compare the distribution of correlation coefficients between repeated target alone presentations with the distribution of correlation coefficients between target alone and target with distractor trials. The authors have done this analysis in rebuttal figure 7 a and b (supplementary figure 11).

Reviewer #3 (Remarks to the Author):

The authors performed several follow-up experiments and analysis to address our concerns.

A primary concern was the verification of the odor dynamics through a PID, in addition to separate control experiments with distractor stimuli alone. Regarding the first, the authors have presented stimulus measurement data corroborating their original findings. These follow-up measurements are an adequate verification. Regarding the second, the authors increased the delay between odor presentations and found that their results stand, even finding contrast enhancement for delayed presentations of the target stimulus. These results fully allay the concerns raised. Finally, the authors satisfactorily addressed through correlation analysis the concerns regarding the statistical analysis, relegating PCA to a more qualitative assessment. This also addresses concerns about trial-to-trial variability using a similar correlation analysis for data within a single trial vs. across trials.

In sum, the careful follow-up experiments and statistical tests corroborate the authors' findings of context-dependending discrimination ability. The revised manuscript and figures are ready for publication.

We thank all three reviewers for their thorough and thoughtful comments that has helped us significantly improve the manuscript. Reviewers 1 and 3 were happy with the previous round of revisions. Reviewer 2 is also satisfied with the new experiments and analysis but has asked for a minor clarification and for inclusion of some results from rebuttal to the manuscript. We have made the necessary changes and hope the manuscript is accepted for publication.

Response to Reviewer 3

I want to note at the beginning that I find the results reported here very interesting and support the publication of this manuscript.

We thank the reviewer for his/her positive evaluation of our work and supporting its publication.

However, my goal is for all the claims reported in the manuscript to be rigorously supported by the data, analyzed appropriately. In the first round of revisions, the authors have adequately addressed almost all the concerns/suggestions I had. I thank them for their transparency and diligence. There are two remaining points though, where I still feel some small changes would improve the manuscript. The changes suggested are small.

We have addressed these small changes with this round of revision as discussed below.

1. The analysis reported in rebuttal figure 2 (Supplementary figure 1 a, b) still fails to quantify the extent to which the data supports the general rule. The authors' claim is justified if in general, the population of blue lines is to the left of the population of red lines. This has not been systematically quantified and in fact, doesn't seem to be the case for hex-app in panel a and ger-benzald in panel b. It should be straightforward to compute means of deviation from baseline for distractors and deviation from normal responses for targets. Then the authors could correlate these two deviations and make a statistically supported claim. This is currently not the case, but looking at the plots, it seems the result of a statistical analysis should be in line with what the authors are claiming.

We have systematically quantified our results using a simple correlation analysis (new Supplementary Fig. 1a, b). We plotted the change in the response evoked by the target odorant when encountered after distractor (with respect to the solitary target odor response) along the y-axis, and the distractor response along the x-axis. Note that zero along the y-axis represents no change in target odor response, positive values indicate increase and negative values indicate reduction in the response to the target odorant when it is sequentially encountered after the distractor. In addition to qualitatively showing the general rule, we have now also computed the correlation between the response change to the target odor and the response to the distractor odorants. As a general rule, these correlations were indeed negative and statistically significant (P value < 0.05).

As noted by the reviewer, most but not all distractor odorants caused a substantial change in the response to the target odorants that were presented after them. Benzaldehyde and apple did not cause much contrast enhancement of the target odor responses. This is evident from

both the PCA trajectories and correlational analysis in Figure 2. Therefore, for these odors the new analysis also did not reveal a significant trend. This is consistent with the data and results presented.

Updated Supplemental Figure 1: (a, b)

- (a) Change in PN responses to the target odorant (y axis) vs. response to distractor (x axis) is plotted for five distractor-target pairs (the target odorant is hex in all cases). The maximum spike rate in a 50 ms time bin during the first 1 s of odor presentation is shown for all PNs that were excited by hex ($n = 36$ PNs for solitary hex; see Methods for PN response categorization). Zero represents identical response to both solitary and sequential presentation of the target odorant. Red lines indicate that the response to target after distractor is less than the response to target alone (i.e. negative values). Blue lines indicate increase in the response for target after distractor when compared to target alone (i.e. positive values). A very small uniform random noise has been added to jitter the points with same x-values and reduce overlap between colored lines. Correlations between the change in target odor response and response to distractor odor (corr values) and their significance levels are shown on each panel.
- (b) Similar plots as panel a but plotted when the target odor was ger ($n = 30$ excitatory PNs to solitary ger).

2. In rebuttal figure 3, the authors show that post-distractor, target responses are sub-linearly predictable from target alone and distractor alone responses. I find this very interesting and am disappointed that it will not be included even as a supplementary figure. Related to this, rebuttal figure 4, also does not make it to the manuscript.

Main figure 2 shows that target responses after distractor presentation are changed from the target alone response. This alone does not imply contrast enhancement. For functionally useful

contrast enhancement, the target odor representation must systematically move away from the distractor representation, while remaining close to the original target representation. The analysis already done by the authors in rebuttal figures 3 and 4 show exactly this! It would be a pity for them to not be included in the publication. I urge the authors to re-consider including these analyses.

These figures have now been included as a new Supplementary Figure (new **Supplementary Fig. 4**)

As an aside, I noticed that the data in rebuttal figure 3 panel b seems identical to main figure 2, panel d. In case the authors decide to include this figure in the supplement, they should rectify this error.

The reviewer is mistaken. Rebuttal Figure 3 panel b is not identical to Main Figure 2 panel d. We think the reviewer is referring to Rebuttal Figure 4 panel b. Even in that case it is not identical to Figure 2d.

This point is not about an un-addressed concern, it is a clarification. On page 15 of the rebuttal, the authors' response highlights a point where I failed to be clear in my original review. Referring to main figure 4, panels e and f, the authors argue that the bias of the correlation coefficient distribution towards 0 rather than 1 indicates changed temporal response patterns. This is the bias I was referring to in my original review. I pointed out that merely having correlation coefficients smaller than 1 does not indicate systematic changes in response patterns in time. Differences between repeat neuronal response measurements to the same stimulus can arise due to a variety of noise sources. One way to address this is to compare the distribution of correlation coefficients between repeated target alone presentations with the distribution of correlation coefficients between target alone and target with distractor trials. The authors have done this analysis in rebuttal figure 7a and b (supplementary figure 11).

Agreed. Thanks for making this suggestion. We believe our results are stronger with the revisions made.